



# Numerical investigation on measurement errors of mixing states of fractal black carbon aerosols using single-particle soot photometer and the effects on radiative forcing estimation

Jia Liu[1, 2, 3], Guang-ya Wang[1, 2, 3], Can-can Zhu[1, 2, 3], Dong-hui Zhou[1, 2, 3], and Lin Wang[1, 2, 3]

[1]Non-destructive Testing Laboratory, School of Quality and Technical Supervision, Hebei University, Baoding, 071002, China.
[2]National & Local Joint Engineering Research Center of Metrology Instrument and System, Baoding, 071002, China.
[3]Hebei Key Laboratory of Energy Metering and Safety Testing Technology, Baoding, 071002, China.

*Correspondence*: Jia Liu (liujia@hbu.edu.cn)

**Abstract.** The mixing state of black carbon (BC) aerosols can be measured by the single-particle soot photometer (SP2). However, the measured mixing state contains errors, because the core-shell model and Mie scattering calculation are employed in the measurement principle of SP2, and the spherical core-shell structure seriously deviated the real morphology of coated BC. In this study, fractal models are constructed to represent thinly and heavily coated BC particles for optical simulations, the scattering cross-section are selected as reference to conduct optical retrieval of particle diameter ($D_p$) based

on Mie theory, just like the measurement principle of SP2, and the diameter of BC core ($D_c$) are the same for fractal and spherical models. Then, the measurement errors of mixing state ($D_p/D_c$) of BC are investigated from numerical aspect, and the estimation accuracy of BC radiative forcing is analyzed through the simple forcing efficiency (SFE) equation with SP2 measurement results taken into consideration. Results show that SP2 measured $D_p/D_c$ based on Mie theory underestimates the realistic mixing state of coated BC for most particle sizes, and the largest relative error for single-particle can be about

42%. The retrieval errors of mixing state of thinly coated BC for both single-particle and particle groups are larger than these of heavily coated BC. In addition, evaluation errors of radiative forcing of coated BC caused by measurement errors of SP2 are up to about 76% and 43% at 1064 and 532 nm, respectively. This study provides meaningful referential understandings of the measured $D_p/D_c$ of SP2.

## 1. Introduction

Black carbon (BC) produced from the incomplete combustion of biomass and fossil fuels is considered to be the second most important factor affecting global warming after carbon dioxide (Zhang et al., 2021). Black carbon aerosols directly affect the climate by absorbing shortwave solar radiation to heat the atmosphere, and also indirectly affect the climate through the complex interaction with clouds (Zhao et al., 2022). The bare BC just emitted from the source has poor hygroscopicity, making it difficult to form cloud condensation nuclei (CCN) (Zhuang et al., 2013). When the hydrophobic BC is mixed with

hydrophilic aerosol during the aging process, the mixed BC aerosol also becomes hydrophilic, which further act as CCN and



participates in microphysical processes of clouds, thus affecting the effective radius and number density of cloud droplets, cloud volume and lifetime, and ultimately the climate (Ching et al., 2016). When deposited to the surface of snow and ice, BC further reduces the snow albedo and accelerates the melting process (Jacobson, 2004). In addition, BC aerosols also affect the air pollution process by altering the thermal structure of the planetary boundary layer, which further affects the

vertical diffusion of air pollutants, enhances haze events, harms human health, and reduces atmospheric visibility (Huang et al., 2018).

The non-spherical fractal morphology of BC has attracted extensive attention. Freshly emitted black carbon particles are chain-like aggregates consisting of a large number of near-spherical monomers. During the aging process in the atmospheric environment, BC will be coated by other species, and their aggregate morphology tends to be more compact (China et al.,

2013). The aging process causes variations in the mixing states of BC particles and results in significant changes in the optical parameters including absorption properties, which in turn affects the radiative forcing of BC and brings great uncertainties to the assessment of the climate effects of BC (Wu et al., 2018; Zeng et al., 2019). Therefore, the measurement and monitoring of the mixing states of BC particles have drawn much attention in the field of atmospheric aerosol observation. Currently, the mixing states of BC aerosols are mainly characterized using two methods: the particle diameter

ratio of the whole particles to the BC core ($D_p/D_c$) and the mass ratio of the coating material and the BC core ($M_R$). Single-particle soot photometer (SP2) can effectively measure the mixing state of BC aerosol and has been widely used in field and laboratory observations. Liu et al. (2014)employed SP2 to measure the particle size and mixing state of BC aerosols in London during wintertime and analyzed their emission sources, results showed that the $D_p/D_c$ of BC particles emitted from three types of sources in the west, southeast, and east were 1.28, 1.45 and 1.69, respectively. Liu et al. (2022) measured $M_R$

under different time backgrounds with the assistance of the tandem system of a Couette centrifugal particle mass analyzer (CPMA) and an SP2 and verified that the mixing state of BC was affected by both relative humidity (RH) and pollutant concentrations. Zhang et al. (2020) employed SP2 to characterize fresh BC particles emitted from four different emission sources and found that coatings are often already present in freshly emitted BC particles, and diesel vehicles emit BC particles with the least coatings ($D_p/D_c$~1.25), while crop residues emit BC particles with the most coatings ($D_p/D_c$~1.75).

The application of SP2 has remarkably improved the understanding of the mixing state of freshly emitted and aged BC particles.

The single-particle soot photometer measures the mass and mixing state of a single BC particle based on the combination of laser-induced incandescent light technology and light scattering measurement technology. In the optical cavity, when a BC particle vertically passes through the high-energy laser beam with a wavelength of 1064 nm, the scattering cross-section of

the BC particle can be rapidly retrieved based on the measurement results of the scattering signal detectors (Schwarz et al., 2006). The coated particles rapidly absorb the laser energy, the coating is heated to vaporization first, and then the refractory BC (rBC) is heated and emits incandescent light (Zhao et al., 2021). The intensity of the incandescent light signal is proportional to the mass of rBC, and the particle diameter of rBC can be obtained based on the preset density ($1.80g/cm^3$). Then, the diameter of the scattering equivalent sphere of coated BC particle can be retrieved based on the Mie scattering



theory with the basic assumption that the coated BC particle has a concentric core-shell structure consisting coating sphere
and BC sphere (Kompalli et al., 2021). Finally, the particle size ratio of the whole particle to the BC core ($D_p/D_c$) can be
obtained. According to the measurement principle SP2, it can be deduced that there are unavoidable measurement errors in
$D_p/D_c$ because the core-shell model used in the retrieval of optical equivalent particle size $D_p$ does not conform to the non-
spherical complex morphology of the coated BC particles. At present, the measurement error of BC mixing state $D_p/D_c$ using
SP2 is difficult to be quantified directly through experimental investigations. Nevertheless, the rapid developments of both
morphology modeling and optical simulation of coated BC particles provide an investigative strategy for evaluating the
measurement accuracy of BC mixing states.

Based on the microscopic morphology of coated BC, a series of models have been constructed in previous studies to
investigate the optical properties of BC. Wu et al. (2014) constructed a fractal closed-cell model with concentric bilayer
spheres as monomers, the optical properties at 450~1000 nm were calculated using the multiple-sphere $T$-matrix method
(MSTM) and found that the optical properties of the closed-cell model differ significantly from the calculated results of the
core-shell model based on Mie scattering theory. Zeng et al. (2019) developed the coated-aggregate model, the calculated
results using the MTSM show that extinction of coated BC is significantly enhanced due to the increase in scattering, and the
enhancement depends on both the amount and hydrophilicity of the coatings. Kanngiesser and Kahnert (2018) designed a
tunable model for the transition from film-coating to spherical-shell coating, optical calculations using the discrete dipole
approximation (DDA) at 355 and 532 nm showed that the rapid and slow transitions of the coating have different effects on
the scattering cross-section, and the overlapping phenomenon ($C_{ov}$) between the BC monomers can also affect the scattering
cross-section. In short, abundant models and numerical simulation algorithms of BC provide convenience for accurate
calculation of BC optical properties and also create an effective way to quantify the possible errors of the mixing states of
BC measured by SP2.

In this study, numerical simulations are performed from the perspective of quantifying the measurement error of $D_p/D_c$ based
on the observed wavelength and complex refractive index consistent with SP2. The typical fractal model of BC particles
with preset volume fractions of BC and coating are constructed, and the optical simulation results of scattering cross section
are regarded as reference values of realistic particles. Then, the volume equivalent sphere of BC in the fractal model is
employed as BC core of the core-shell model, and the optical equivalent particle diameter of the coated BC is retrieved based
on the scattering cross-section of fractal model using Mie scattering theory, which is similar to the measurement principle of
SP2. The retrieved value of $D_p/D_c$ of optically equivalent core-shell model can be obtained. Finally, comparative analyses of
the differences between the retrieved $D_p/D_c$ of the core-shell model and the preset $D_{p,v}/D_{c,v}$ of the fractal model enable the
evaluation of the SP2 measurement accuracy of BC particle mixing states from numerical aspects.

Furthermore, the effect of measurement errors of mixing states on the radiative forcing evaluation of BC is discussed using
the simple forcing efficiency equation to provide insight of the possible errors when SP2 measurement results are employed
in climate modes.



## 2. Methodology

### 2.1 Model construction of BC aerosols

According to observations of electron microscope, the morphology of freshly emitted black carbon particles often appears as chain-like aggregates. The geometry of the bare BC can be constructed according to the well-known fractal aggregate framework, and the mathematical description is as follows (Sorensen, 2001):

$$N_s = k_f \left( \frac{R_g}{a_0} \right)^{D_f} \quad , \tag{1}$$

$$R_g = \sqrt{\frac{1}{N} \sum_{i=1}^{N} r_i^2} \quad , \tag{2}$$

where $a_0$ is the radius of monomers, $N_s$ is the number of monomers, $k_f$ is the fractal prefactor and $D_f$ is the fractal dimension that control the morphology of BC aggregates. $R_g$ is the radius of gyration which measures the spatial size of the aggregate, $r_i$ is defined as the distance between the $i$th monomer and the mass center of the whole aggregate.

During the aging process in the atmosphere, the surface of BC will be covered by organics and inorganic salt, forming complex coating structures (Kholghy et al., 2013). In order to more accurately simulate BC particles, two fractal aggregate models that are more suitable for the actual morphology of BC are selected for numerical simulation, as shown in **Figure 1(a,**
**b)**, the closed-cell and coated-aggregate models stand for thinly coated BC and heavily coated BC, respectively. The mixing state is represented by the ratio of the volume equivalent sphere diameter of coated particle to that of BC core ($D_{p,v}/D_{c,v}$). In addition, as shown in **Figure 1(c)**, the core-shell model based on the spherical assumption and applicable for Mie scattering theory is constructed, and mixing state can be directly expressed by the ratio $D_p/D_c$. For the construction of closed-cell and
coated-aggregate models, the tunable diffusion-limited aggregation (DLA) software developed by Wozniak et al. (2012) are used at first to generate bare BC fractal aggregate, then spherical coatings were added based on original aggregate (Liu et al., 2023). The ratio $D_{p,v}/D_{c,v}$ can be obtained from the volume fraction of BC core ($V_f$) according to the mathematical formula shown below:

$$V_f = \frac{V_{BC}}{V_{total}} \quad , \tag{3}$$

$$\frac{D_{p,v}}{D_{c,v}} = \frac{1}{\sqrt[3]{V_f}} \quad , \tag{4}$$

$$D_{c,v} = 2a_0 \sqrt[3]{N_s} \quad , \tag{5}$$

where $V_{BC}$ and $V_{total}$ are the volume of BC core and the whole coated BC particle, respectively.



**Figure 1.** Geometries of fractal BC aggregate model with $N_s$=100 and core-shell model. **(a)** Closed-cell model with $D_f$=2.4

and $V_f$=0.40. **(b)** Coated-aggregate model with $D_f$=2.6 and $V_f$=0.10. **(c)** Core-shell model with $V_f$=0.40.

In this study, the value of fractal prefactor ($k_f$) is assumed to be 1.20 (Sorensen and Roberts, 1997), and the radius of BC

monomer ($a_0$) is fixed at 20 nm which is a typical value observed in experiments (Li et al., 2003). Meanwhile, $N_s$ range from

50 to 2000 with a step size of 50, which covers the $D_c$ measured by SP2. The remaining microphysical parameters $D_f$ and $V_f$

also affect the morphology of the fractal aggregate model, and the corresponding relationship between $D_{p,v}/D_{c,v}$ and $V_f$ are

shown in **Table 1**.

**Table 1.** Morphological descriptors of BC fractal aggregate model.

**2.2 Numerical simulation of optical properties**

Many methods have been used to calculate the optical properties of BC fractal aggregates, such as the Rayleigh-Debye-Gans

(RDG) approximation, the discrete dipole approximation (DDA), and the *T*-matrix method (Adachi et al., 2010; Li et al.,

2016; Mishchenko et al., 2013). As one of the most computationally efficient and accurate methods, the multiple-sphere *T*-

matrix method (MSTM) is developed based on *T*-matrix theory and employs the addition theorem of vector spherical wave

functions to account for the interactions between different spherical monomers in multi-sphere systems (Mackowski, 2014).

The MSTM code has been widely used in numerical simulation studies to calculate the optical properties of BC fractal

aggregates (He et al., 2015). The wavelength of the optical property simulation is set to 1064 nm in this study, which is the

same as the laser wavelength used in the SP2. Typically, the refractive indices of BC core and coating materials were

assumed to be 2.26-1.26$i$ and 1.50+0$i$ respectively, and the densities of BC core ($\rho_{BC}$) and coatings ($\rho_{coating}$) were set to 1.80

g/cm$^3$ and 1.20 g/cm$^3$ (Zhang et al., 2019; Liu et al., 2014; Turpin and Lim, 2001). In addition, the selections of the refractive

index and density are consistent for both MSTM calculation and Mie scattering calculation. MSTM code can directly

calculate the scattering efficiency ($Q_{sca}$), extinction efficiency ($Q_{ext}$), absorption efficiency ($Q_{abs}$) and asymmetry parameter

($ASY$) of BC fractal aggregates. According to the measurement principle of SP2, the scattering cross-section ($C_{sca}$) is

calculated for the subsequent retrieval:

$$C_{sca} = Q_{sca} \times \pi \times \left( \frac{D_p}{2} \right)^2 \ , \tag{6}$$

**2.3 Retrieval of mixing state of BC**

For the SP2, the incandescent signals detected can determine the mass of each single BC core, then the diameter of the BC

core ($D_c$) can be deduced based on its density. The leading-edge-only (LEO) fit method developed by Gao et al. (2007)

constructs a complete Gaussian scattering function using the scattering signal of coated BC particles before it is perturbed by



the laser, and the Gaussian scattering function can be used to determine the scattering cross-section ($C_{sca}$) of coated BC particle. Further, the diameter of coated BC particles ($D_p$) can be retrieved using Mie scattering theory. Based on the measurement principle of SP2, the retrieval of mixing state $D_p/D_c$ with the scattering cross-section as a reference is the key

of this study.

The specific retrieval process is shown in **Figure 2**: Firstly, the optical property ($C_{sca}$) of the BC fractal aggregate models with preset $D_{p,v}/D_{c,v}$ at 1064 nm are calculated based on MSTM code. Secondly, the optical properties of the core-shell model are calculated using Mie scattering theory, and the value of $D_c$ is the same as the $D_{c,v}$. Lastly, the core-shell model whose scattering cross-section has the smallest difference from that of fractal aggregate model is retrieved, and its $D_p/D_c$ is

regarded as the measured mixing state of coated BC particle. Furthermore, a simplified conceptual retrieval method based on ordinary least squares was defined as follows:

$$\chi^2 = \left[ \frac{C_{sca-MSTM}\left(D_{p,v}\right) - C_{sca-Mie}\left(D_p\right)}{C_{sca-MSTM}\left(D_{p,v}\right)} \right]^2 , \tag{7}$$

where $C_{sca\text{-}MSTM}$ and $C_{sca\text{-}Mie}$ are the simulated scattering cross-sections of BC fractal aggregate models and core-shell models, respectively. In addition, the relative error ($RE$) is used to represent the mixing state retrieval error of BC.

$$RE = \frac{D_p/D_c - D_{p,v}/D_{c,v}}{D_{p,v}/D_{c,v}} \times 100\% , \tag{8}$$

**Figure 2** also shows the verification process when SP2 measurements are used to predict the optical properties of coated BC at 532 nm. The optical properties of BC fractal aggregate models with the preset particle size ($D_{p,v}$) and the core-shell model with the retrieved particle size ($D_p$) at 532 nm are calculated based on the MSTM code and Mie scattering theory, respectively. The discussion of calculated normalized scattering and absorption properties is shown in section 3.3.


**Figure 2**. Schematic overview of the methodology of both BC mixing state retrieval and the verification of optical property at 532 nm predicted based on SP2 measurement results.

### 3. Result and discussion

### 3.1 Effect of BC aerosol morphology on the retrieval of mixing state

**Figure 3** shows the retrieval results and relative errors ($RE$) of mixing states of thinly coated soot under different fractal dimensions ($D_f$) and volume equivalent particle diameter ratio shell/core ($D_{p,v}/D_{c,v}$), the colored lines and bars stand for retrieved $D_p/D_c$ and $RE$, respectively. The black dashed line in each figure is the preset value of $D_{p,v}/D_{c,v}$, and the retrieved result is the $D_p/D_c$ with the smallest relative error to the preset $D_{p,v}/D_{c,v}$. Since the shell must be larger than the core of the core-shell model, the orange dashed line in **Figure 3(d)** indicates the lower bound for the retrieval ($D_p/D_c=1$). For the thinly





coated soot, the relative errors of retrieved $D_p/D_c$ are almost always negative, which indicates that SP2 underestimates $D_p/D_c$ when measuring the mixing state of slightly aged black carbon. As shown in **Figure 3**, the retrieved results and relative errors of thinly coated closed-cell models with different fractal dimensions have a similar variation trend with the increase of the volume equivalent diameter of soot core, both retrieved results and relative errors first decrease and then increase, and retrieved results tend to the preset volume equivalent particle core/shell ratio as particle size constantly increases. In addition,

the retrieved results become smaller as the fractal dimension decreases. It should be noted that when both the preset values of the core/shell ratio and the fractal dimension are small, some exceptions in the retrieved results of mixing state of soot appear. As can be seen from **Figure 3(d)**, there are no retrieved $D_p/D_c$ of closed-cell models with fractal dimension of 1.80 at all, and only partial results of $D_p/D_c$ can be obtained for thinly coated soot with fractal dimension of 2.40 and 2.60, which means that the measured $D_p/D_c$ may lose data of large amount of slightly aged atmospheric BC and thus the measured

mixing states of BC in previous field campaigns are larger than the real values. This phenomenon is due to the large difference in morphology between the thinly coated closed-cell model and the core-shell model, resulting in large differences in the scattering cross-section. As clearly shown in **Figure 4**, the retrieved mixing states of heavily coated BC particles are more closed to preset values than thinly coated BC particles, the relative errors of retrieved results are mostly positive and smaller than 10%, indicating that the SP2 measurements slightly overestimate the $D_p/D_c$ of the severely aged BC particles,

and the variation of retrieved results with volume equivalent diameter of BC core are similar for particles with different dimensions. In addition, there are no missed retrieved results ($D_p/D_c$ less than 1) in the considered particle size range, indicating that the measured mixing state of heavily coated atmospheric BC particles are more accurate than that of thinly coated BC particles.

**Figure 3**. Retrieved mixing state ($D_p/D_c$) and relative error ($RE$) as functions of volume equivalent diameter ($D_{c,v}$) for thinly coated BC particles with different $D_f$ and $D_{p,v}/D_{c,v}$. The colored lines stand for retrieved $D_p/D_c$ and the colored bars stand for $RE$.

**Figure 4**. Retrieved mixing state ($D_p/D_c$) and relative error ($RE$) as functions of volume equivalent diameter ($D_{c,v}$) for heavily coated BC particles with different $D_f$ and $D_{p,v}/D_{c,v}$. The colored lines stand for retrieved $D_p/D_c$ and the colored bars stand for $RE$. (a) Coated-aggregates with $D_f = 2.60$; (b) Coated-aggregates with $D_f = 2.80$.

The distribution of retrieved results of mixing states for single-particle with different fractal dimensions over the entire particle size range is shown in **Figure 5**, and the filling width represents the probability distribution of retrieved $D_p/D_c$. As mentioned previously, with $D_{p,v}/D_{c,v}$ and $D_f$ decrease, there will be no retrieved results for some particle sizes or even among

the whole particle size range. The last one data point in **Figure 5(a)** and the last two data points in **Figure 5(b)** are much smaller due to the reduced amount of retrieved results. The retrieved results for $D_{p,v}/D_{c,v}=1.13$ even do not show in **Figure 5(a)** because the retrieval cannot be performed. Therefore, the particle mixing states which are lack of retrieved results are not considered in the following adequacy verification when SP2 measurements are employed to predict the optical properties and radiative forcing of aged soot aerosols.



It can be seen from the figures that the probability distribution of retrieved mixing states are similar for particles employing the same model. The retrieved $D_p/D_c$ for soot with more compact structures have higher retrieval accuracy, and the distributions are more uniform. As for the comparison of these two morphological models for soot with different aging state, the retrieved $D_p/D_c$ of coated-aggregate model is more concentrated around the preset values, which confirms that the mixing state measurement error of SP2 is relatively smaller for heavily coated BC particles.


**Figure 5**. The distributions of retrieved $D_p/D_c$ of different models with different fractal dimensions were demonstrated separately. (a) thinly coated BC with $D_f$ =2.40; (b) thinly coated BC with $D_f$ =2.60; (c) heavily coated BC with $D_f$ =2.60; (d) heavily coated BC with $D_f$ =2.80. The box represents the 25th and the 75th percentiles, the whisker represents the maximum and minimum values of the retrieved results after the removal of outliers with large deviations, and the line in box represents
the average value of retrieved $D_p/D_c$.

**3.2 Retrieved mixing state for particles with a certain size distribution**

It is known from previous observations and experiments that the bulk optical properties of realistic atmospheric BC particles are integrated over a certain particle size distribution. In order to better understand the accuracy of SP2 on the $D_p/D_c$ measurements of the bulk BC particles, the retrieved results for particle groups are also considered in this study. It is
assumed that the equivalent volume diameters of coated BC particles follow the log-normal distribution:

$$n(d) = \frac{1}{\sqrt{2\pi} d \ln(\sigma_g)} \exp\left[ -\left( \frac{\ln(d) - \ln(d_g)}{\sqrt{2} \ln(\sigma_g)} \right)^2 \right] , \qquad (9)$$

where $\sigma_g$ is the geometric standard deviation and $d_g$ is the geometric mean diameter. $d$ is the diameter of the sphere with the same volume as that of the whole coated BC aggregates (i.e., volume equivalent sphere). Particle size distribution of coated BC aggregates with $d_g$=0.15 μm (Yu and Luo, 2009) and $\sigma_g$=1.59 (Zhang et al., 2012) is considered, and these BC particles
have the same preset $D_p/D_c$. The retrieved $D_p/D_c$ and relative errors of the BC particles with size distribution mentioned above are shown in **Table 2**. The relative error of retrieved results for the closed-cell model decreases with the increase of fractal dimension, due to the fact that the structure gradually becomes compact and close to the core-shell model. However, the relative errors of retrieved results of particles with the same $D_f$ are close to each other. In addition, the relative errors for the closed-cell model are larger than those of the coated-aggregate model, because the coated-aggregate model is much
closer to the core-shell model in shape.

**Table 2.** Retrieved $D_p/D_c$ and relative errors of BC particle groups

**3.3 Verification of adequacy when SP2 measurements are employed to predict optical properties of BC**

The optical equivalent diameter $D_p$ and mixing state ($D_p/D_c$) can be measured by SP2. Then, the optical properties of black carbon particles except for scattering properties or even optical properties at other wavelengths can be predicted based on





Mie scattering calculation of core-shell model. In this study, the prediction accuracy of optical properties at 532 nm are selected as typical example. The optical simulation results of the fractal particle models are used to represent the optical properties of the actual atmospheric BC particles, while the Mie scattering results of the core-shell model represent the predicted optical properties of the BC particles inferred from the SP2 measurements. The comparisons of the scattering cross-section are shown in **Figure 6**, it can be found that optical properties of BC particles with different models have

obvious differences. There are two fundamental reasons for these differences: the first is that different structures of these three models (core-shell model, coated-aggregate model and closed-cell model) consequentially result in differences in optical properties, the second is that the optical equivalent core-shell models retrieved based on SP2 measurements have distinct diameters to the preset diameters of both coated-aggregate models and closed-cell models. The distinctions of the scattering cross-sections between the closed-cell model and the core-shell model are small when particle sizes and preset

$D_p/D_c$ is small, and the distinctions generally increase with the particle diameter, the relative errors range between 40% and 80%. As shown in **Figure 6(a-b)**, the numerical simulation results of the closed-cell model are larger than those of the core-shell model, which indicates that the scattering cross-section of thinly coated BC particles at 532 nm predicted based on the measurement results of SP2 at 1064 nm will be obviously underestimated when compared to their realistic scattering cross-section. However, **Figure 6(c-d)** illustrates that the opposite is true for heavily coated BC particles and the scattering

properties predicted using core-shell model is smaller than the realistic properties of coated-aggregate model at 532 nm. Furthermore, thinly coated BC particles with $D_{p,v}/D_{c,v}$=2.15 (the thinnest coating) and heavily coated BC particles with $D_{p,v}/D_{c,v}$=2.71 (the heaviest coating) have the best predict accuracies among these considered mixing state.

**Figure 6**. Differences in scattering cross-section between fractal particle model (solid lines) and core-shell model (dashed

lines) at 532 nm. **(a)** Closed-cell model with $D_f$ =2.40; **(b)** Closed-cell model with $D_f$ =2.60; **(c)** Coated-aggregate model with $D_f$ =2.60; **(d)** Coated-aggregate model with $D_f$ =2.80.

In addition, the radiative forcing of BC aerosol is closely related to the absorption and scattering properties. In order to investigate the effect of measurement errors of SP2 caused by morphological model selection on the estimation of BC radiation effect, optical properties mass scattering cross-section (MSC) and mass absorption cross-section (MAC) are

calculated for further investigation:

$$MAC = \frac{C_{sca}}{m_{total}} \ , \tag{10}$$

$$MSC = \frac{C_{abs}}{m_{total}} \ , \tag{11}$$

$$m_{total} = \rho_{BC}V_{BC} + \rho_{coating}V_{coating} \ , \tag{12}$$

$$V_{coating} = \left(1 - V_f\right)V_{BC}\Big/V_f \ , \tag{13}$$





where $m_{total}$ is the mass of whole coated BC particle and $V_{coating}$ is the volume of coating material. The values of density of

both BC and coating are mentioned in Section 2.2.

**Figure 7** and **Figure 8** demonstrate the mass absorption cross-section and mass scattering cross-section of fractal models

and core-shell models, respectively. The mass absorption cross-section of the closed-cell model slightly increases with the

increase of both $D_{p,v}/D_{c,v}$ and $D_f$, but the size of BC core has little influence. The Mie theory calculation results of the optical

equivalent core-shell model based on SP2 measurements increase with the increase of $D_{p,v}/D_{c,v}$ for small particles, and the

influence of $D_{p,v}/D_{c,v}$ disappears when the BC core size increases to about 320 nm. Furthermore, the predicted MAC of small

particles is larger than the actual MAC of thinly coated BC, and when the size of BC core increases to certain values in the

range of 210-240 nm, the predicted MAC equates to the actual MAC, after that the opposite is always true for large particles.

As shown in **Figure 7 (c-d)**, the fractal dimension and mixing state have small effect on the mass absorption cross-section

on the coated-aggregate model, but the differences in MAC between the coated-aggregate model and the core-shell model is

apparent. This indicates that the predicted results of mass absorption cross-section deviate significantly from the actual

values of the heavily coated BC particles, and the maximum relative error is up to about 60%. **Figure 8** demonstrates that the

mass scattering cross-sections of the closed-cell model increases with particle size, fractal dimension and mixing state, and

they are obviously larger than the corresponding results of the core-shell model in most cases over the entire particle size

range, indicating that the predicted results of MSC based on SP2 measurements underestimate the mass scattering cross

section of coated BC at 532 nm wavelength. In contrast, the mass scattering cross-sections of both the coated-aggregate

model and the core-shell model decrease with the increase of particle size, as shown in **Figure 8(c-d)**, but the predicted MSC

overestimates the mass scattering cross-section of BC over most of the particle size range, and it is hard to parameterize the

errors of the predicted results.


**Figure 7**. Differences in mass absorption cross-section (MAC) between fractal particle model (solid lines) and core-shell
model (dashed lines) at 532 nm. **(a)** Closed-cell model with $D_f$ =2.40; **(b)** Closed-cell model with $D_f$ =2.60; **(c)** Coated-
aggregate model with $D_f$ =2.60; **(d)** Coated-aggregate model with $D_f$ =2.80.

**Figure 8**. Similar to **Figure 7**, but the solid and dashed lines represent the mass scattering cross-section (MSC) of fractal
particle model and core-shell model, respectively.

**3.4 Effects of SP2 measurement errors on the estimation of BC radiative forcing**

The microphysical properties of black carbon (morphological structure, size distribution, and mixing state) have a significant

impact on their optical properties, which further affect the radiative effect of black carbon. Lu et al. (2020) revealed that the

uncertainty of the radiative forcing and heating rate of BC due to different geometries and size distributions of BC is less

than 30%, while the uncertainty due to the mixing state of BC is as high as 80%. SP2 measurement the mixing state of BC

based on the Mie scattering theory and BC particles are assumed to have spherical core-shell structures. But the core-shell





model is seriously out of line with the actual morphology of BC, so the measured mixing state have unavoidable errors, which in turn affect the estimation of the radiative effect of BC. In this study, the simple forcing efficiency (SFE) equation

developed by Bond and Bergstrom (2006) is employed to quantify the radiative forcing evaluation errors caused by mixing state measurement errors of SP2. The SFE is defined as radiative forcing normalized by BC mass and represents the energy added to the Earth's atmospheric system by a given mass of particles in the atmosphere:

$$\frac{dSFE}{d\lambda} = -\frac{1}{4}\frac{dS(\lambda)}{d(\lambda)}\tau^2\left(1-F_c\right)\left[2\left(1-a_s\right)^2\beta(\lambda)\cdot MSC(\lambda) - 4a_s MAC(\lambda)\right]\,,\tag{14}$$

where $\dfrac{\mathrm{d}S(\lambda)}{\mathrm{d}(\lambda)}$ is the spectral solar irradiance according to ASTM G173-03, the atmospheric transmittance $\tau$=0.79, the cloud

fraction $F_c$=0.6, typical urban surface albedo $a_s$=0.19, and the backscatter fraction$\beta$=0.15.

**Table 3** and **Table 4** show the calculated results of SFE at 532 nm and 1064 nm, and it can be found that all the values of radiative forcing of coated BC particles are positive. The "actual value" is the SFE of actual thinly and heavily coated BC particles represented by closed-cell model and coated-aggregate model, the optical properties are obtained using MSTM. The "measured value" is the SFE obtained using recalculated MSC and MAC based on the optical equivalent core-shell models

from SP2 measurement results. The effect of SP2 measurement error on SFE are disparate at different wavelengths, the relative errors range from 23% to 76% at 1064 nm and range from -10% to 43% at 532 nm. In addition, the distinctions of SFE between the closed-cell model and the corresponding optical equivalent core-shell model is smaller than the distinctions of SFE between the coated-aggregate model and the core-shell model at 1064 nm, which indicates that the SP2 measurement error has more significant influence on the radiative forcing of the heavily coated BC. At 532 nm, the relative error of SEF

for closed-cell model is larger than coated-aggregate model for small $D_{p,v}/D_{c,v}$, but the opposite is true when $D_{p,v}/D_{c,v}$=2.71. Even though the SFEs of optical equivalent core-shell models corresponding to the heavily coated BC are almost equal for particles with different fractal dimensions, the SFE of the coated-aggregate model differs greatly in different fractal dimensions, leading to various relative errors between "actual value" and "measured value". The SFE of the closed-cell model increases with the fractal dimension and the mixing state, and the relative error between closed-cell model and the

corresponding core-shell model also follow the same trend. It should be noted that the effect of the SP2 measurement error on the estimation of the radiative forcing of thinly coated BC cannot be ignored.

**Table 3.** The SFE of both the fractal soot models and the SP2 retrieved core-shell models at 1064 nm.

**Table 4.** The SFE of both the fractal soot models and the SP2 retrieved core-shell models at 532 nm.



## 4. Conclusions

BC aerosols have strong light absorption capacity compared with other atmospheric aerosols, and they can disturb the regional and global radiation balance and produce positive radiative forcing by absorbing solar radiation, which eventually leads to global warming. The content and structure of coating significantly change the mixing state ($D_p/D_c$) of black carbon particles, and the lens effect caused by coating has important and complicated effects on the optical properties of black carbon, which brings great uncertainty to the radiative effect and ultimately affects the accurate assessment of the climate effect of black carbon aerosols.

Single-particle soot photometer (SP2) employs the spherical core-shell model to represent coated BC particles, but this inappropriate model introduces errors to the $D_p/D_c$ measurement results and is not conducive to accurately obtaining the mixing state of coated fractal BC particles in the atmosphere. This study quantifies the SP2 measurement error of the mixing state of coated BC particles using typical fractal morphological models of both thinly and heavily coated BC from the perspective of numerical simulations and evaluates the effect of measurement error on the estimation of radiative forcing of BC aerosols. The main conclusions are summarized as follows:

1. The mixing state of BC particles measured by SP2 is underestimated compared with the realistic $D_p/D_c$. Relative error in the measured mixing state of single BC particles under different coating conditions is significantly different. The relative error of thinly coated BC particle represented by the closed-cell model is larger than that of heavily coated BC particle represented by the coated-aggregate model, and the maximum relative error is about 42%. With the decrease of fractal dimension, the relative errors of the mixing state of thinly coated BC particles with loose structures increase.

2. For BC groups whose particle size follows a certain log-normal distribution, the SP2 measurements of $D_p/D_c$ will be underestimated. The relative errors of $D_p/D_c$ for closed-cell model are larger than that for coated-aggregate model, indicating that the SP2 measurement accuracy of the mixing state of heavily coated BC particles is better than that of thinly coated BC particles.

3. When the $D_p/D_c$ measured by SP2 are used to predict optical properties of coated BC, it can be found that the predicted mass absorption cross-section (MAC) and mass scattering cross-section (MSC) at 532 nm recalculated based on core-shell model have significant discrepancies with that of realistic thinly and heavily coated BC particles. The largest difference between the predicted MAC and the reference value is up to about 60%.

4. The measurement errors of mixing state have larger effect on the estimation accuracy of radiative forcing for heavily coated BC particles than that for thinly coated BC particles at both 1064 and 532 nm wavelengths. Furthermore, the relative error of estimated radiative forcing of heavily coated BC particles reaches about 76% at 1064 nm, while it reaches about 43% at 532 nm.

*Data availability.* The data for this study is available online (https://doi.org/10.5281/zenodo.7589824).



*Author contributions*. **Jia Liu:** Conceptualization, Supervision, Writing-review & editing; **Guang-ya Wang:** Investigation, Formal analysis, Writing-original draft; **Can-can Zhu:** Formal analysis, Writing-review & editing; **Dong-hui Zhou:** Data curation, Writing-review & editing; **Lin Wang:** Data curation, Formal analysis, Writing-review & editing.

*Competing interests*. The authors declare that they have no conflict of interest.

*Acknowledgments*. We particularly thank Dr. Mishchenko M. I. and Dr. Mackowski D. W. for the MSTM code. We also appreciate the support of the supercomputing center of Hebei University. This study is financially supported by the Natural Science Foundation of Hebei Province (D2021201002), the Science and Technology Project of Hebei Education Department (QN2021013), the High-level Talents Research Start Project of Hebei University (521000981417).

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





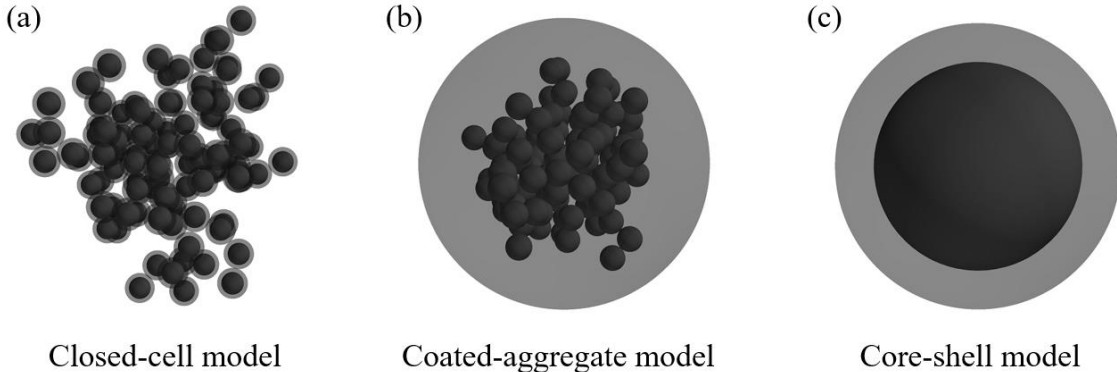

**Figure 1.** Geometries of fractal BC aggregate model with $N_s$=100 and core-shell model. **(a)** Closed-cell model with $D_f$=2.4 and $V_f$=0.40. **(b)** Coated-aggregate model with $D_f$=2.6 and $V_f$=0.10. **(c)** Core-shell model with $V_f$=0.40.

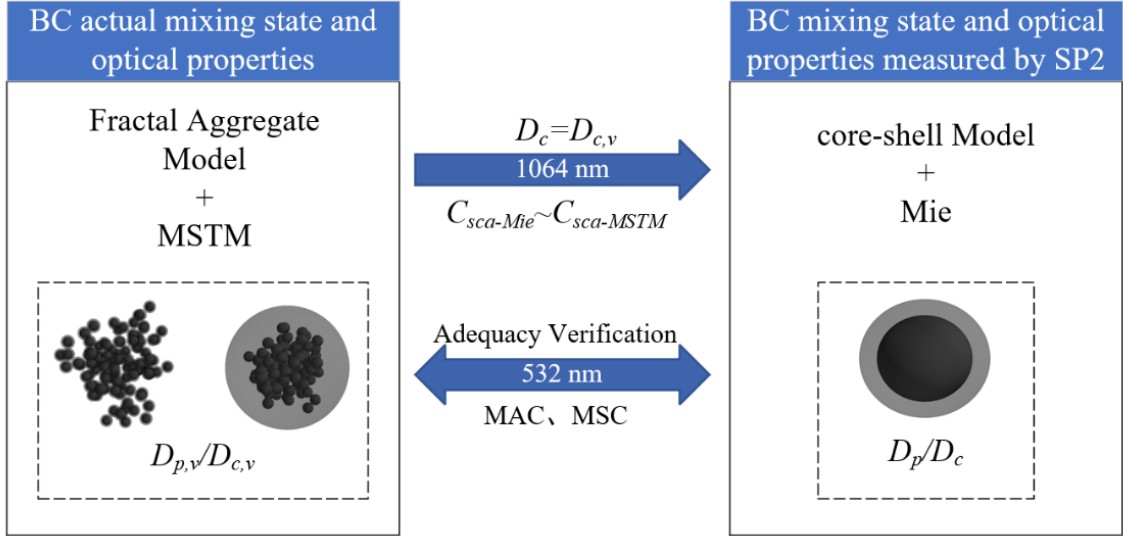

**Figure 2.** Schematic overview of the methodology of both BC mixing state retrieval and the verification of optical property at 532 nm predicted based on SP2 measurement results.



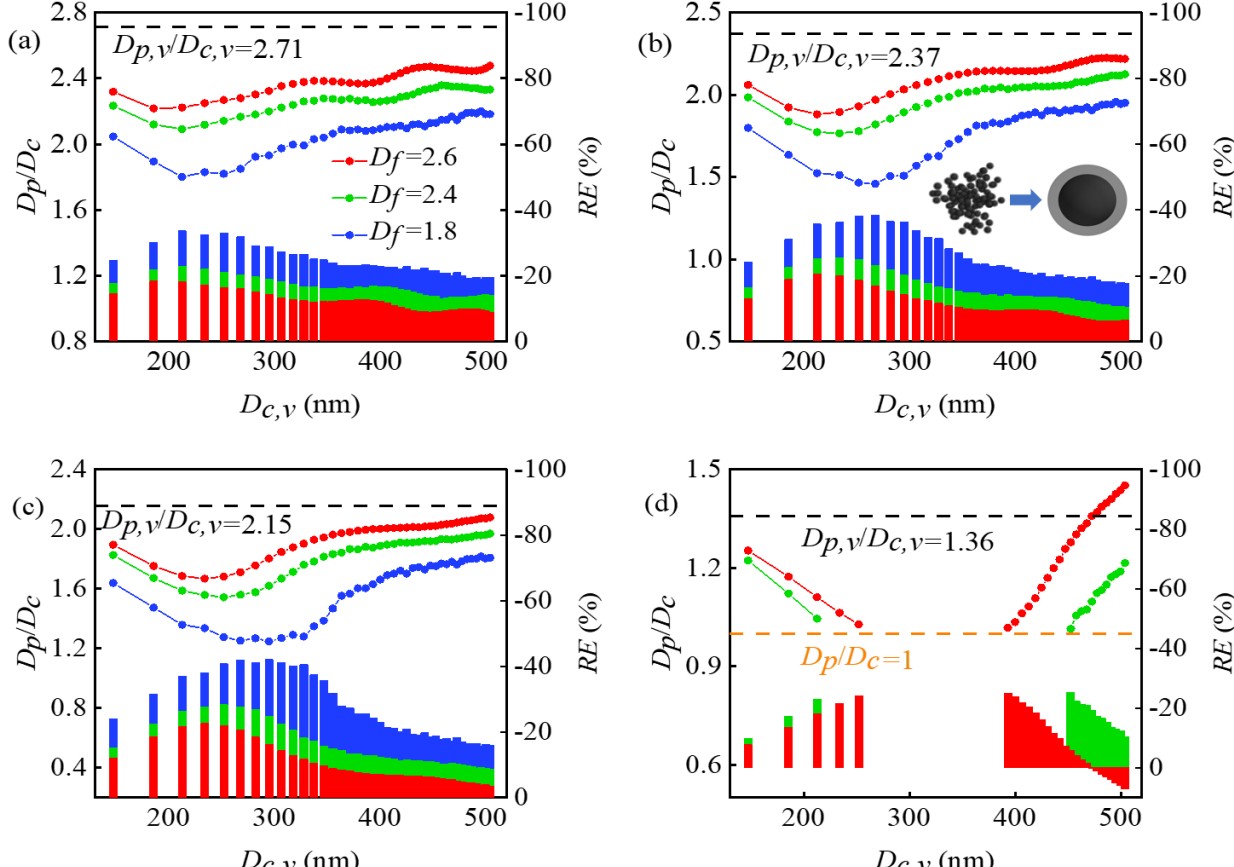

**Figure 3.** Retrieved mixing state ($D_p/D_c$) and relative error ($RE$) as functions of volume equivalent diameter ($D_{c,v}$) for thinly coated BC particles with different $D_f$ and $D_{p,v}/D_{c,v}$. The colored lines stand for retrieved $D_p/D_c$ and the colored bars stand for $RE$.

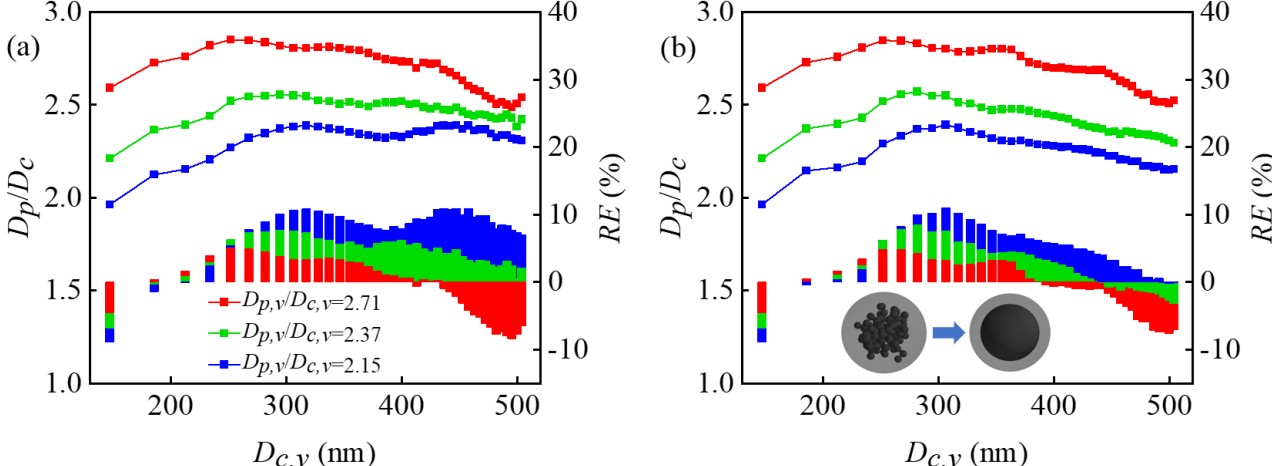

**Figure 4**. Retrieved mixing state ($D_p/D_c$) and relative error ($RE$) as functions of volume equivalent diameter ($D_{c,v}$) for heavily coated BC particles with different $D_f$ and $D_{p,v}/D_{c,v}$. The colored lines stand for retrieved $D_p/D_c$ and the colored bars stand for $RE$. (a) Coated-aggregates with $D_f = 2.60$; (b) Coated-aggregates with $D_f = 2.80$.



**Figure 5.** The distributions of retrieved $D_p/D_c$ of different models with different fractal dimensions were demonstrated separately. (a) thinly coated BC with $D_f$ =2.40; (b) thinly coated BC with $D_f$ =2.60; (c) heavily coated BC with $D_f$ =2.60; (d) heavily coated BC with $D_f$ =2.80. The box represents the 25th and the 75th percentiles, the whisker represents the maximum and minimum values of the retrieved results after the removal of outliers with large deviations, and the line in box represents the average value of retrieved $D_p/D_c$.


**Figure 6**. Differences in scattering cross-section between fractal particle model (solid lines) and core-shell model (dashed lines) at 532 nm. **(a)** Closed-cell model with $D_f$ =2.40; **(b)** Closed-cell model with $D_f$ =2.60; **(c)** Coated-aggregate model with $D_f$ =2.60; **(d)** Coated-aggregate model with $D_f$ =2.80.

**Figure 7**. Differences in mass absorption cross-section (MAC) between fractal particle model (solid lines) and core-shell model (dashed lines) at 532 nm. **(a)** Closed-cell model with $D_f$ =2.40; **(b)** Closed-cell model with $D_f$ =2.60; **(c)** Coated-aggregate model with $D_f$ =2.60; **(d)** Coated-aggregate model with $D_f$ =2.80.




**Figure 8**. Similar to **Figure 7**, but the solid and dashed lines represent the mass scattering cross-section (MSC) of fractal particle model and core-shell model, respectively.





**Table 1.** Morphological descriptors of BC fractal aggregate model.

|  | Parameters | Values |
|---|---|---|
| All | monomer radius ($a_0$) | 20 (nm) |
| All | Monomer numbers ($N_s$) | 50-2000, step length 50 |
| All | Fractal prefactor ($k_f$) | 1.20 |
| Thinly coated | Fractal dimension ($D_f$) | 1.80, 2.40, 2.60 |
|  | Volume fraction ($V_f$) | 0.70, 0.40, 0.10, 0.075, 0.05 |
|  | Volume equivalent diameter ratio of shell/core ($D_{p,v}/D_{c,v}$) | 1.13, 1.36, 2.15, 2.37, 2.71 |
| Heavily coated | Fractal dimension ($D_f$) | 2.60, 2.80 |
|  | Volume fraction ($V_f$) | 0.10, 0.075, 0.05 |
|  | Volume equivalent diameter ratio of shell/core ($D_{p,v}/D_{c,v}$) | 2.15, 2.37, 2.71 |

**Table 2.** Retrieved $D_p/D_c$ and relative errors of BC particle groups

| | closed-cell model | | | | | |
|---|---|---|---|---|---|---|
| | $D_f$ =1.80 | | $D_f$ =2.40 | | $D_f$ =2.60 | |
| preset $D_p/D_c$ | retrieved $D_p/D_c$ | RE | retrieved $D_p/D_c$ | RE | retrieved $D_p/D_c$ | RE |
| 2.71 | 1.94 | -28.4% | 2.18 | -19.6% | 2.28 | -15.9% |
| 2.37 | 1.63 | -31.1% | 1.89 | -20.3% | 1.99 | -16.1% |
| 2.15 | 1.42 | -34.0% | 1.69 | -21.5% | 1.80 | -16.7% |

| | coated-aggregate model | | | | |
|---|---|---|---|---|---|
| | $D_f$ =2.60 | | $D_f$ =2.80 | | |
| preset $D_p/D_c$ | retrieved $D_p/D_c$ | RE | retrieved $D_p/D_c$ | RE | |
| 2.71 | 2.63 | -3.1% | 2.63 | -3.1% | |
| 2.37 | 2.26 | -4.6% | 2.26 | -4.5% | |
| 2.15 | 2.02 | -6.0% | 2.03 | -5.9% | |

**Table 3.** The SFE of both the fractal soot models and the SP2 retrieved core-shell models at 1064 nm.

| $D_{p,v}/D_{c,v}$ | SFE | closed-cell | | | coated-aggregate | |
|---|---|---|---|---|---|---|
| | | $D_f$ =1.80 | $D_f$ =2.40 | $D_f$ =2.60 | $D_f$ =2.60 | $D_f$ =2.80 |
| 2.71 | actual value | 0.15 | 0.15 | 0.16 | 0.19 | 0.20 |
| | measured value | 0.23 | 0.25 | 0.26 | 0.33 | 0.33 |
| | relative error | 53.97% | 64.14% | 65.99% | 72.32% | 64.90% |
| 2.37 | actual value | 0.15 | 0.15 | 0.15 | 0.18 | 0.19 |
| | measured value | 0.20 | 0.23 | 0.24 | 0.30 | 0.30 |
| | relative error | 37.37% | 52.29% | 55.64% | 68.13% | 55.19% |
| 2.15 | actual value | 0.15 | 0.15 | 0.15 | 0.16 | 0.19 |
| | measured value | 0.18 | 0.21 | 0.22 | 0.28 | 0.28 |
| | relative error | 23.09% | 41.07% | 46.15% | 75.64% | 48.98% |





**Table 4.** The SFE of both the fractal soot models and the SP2 retrieved core-shell models at 532 nm.

| $D_{p,v}/D_{c,v}$ | SFE | closed-cell | | | coated-aggregate | |
|---|---|---|---|---|---|---|
| | | $D_f$=1.80 | $D_f$=2.40 | $D_f$=2.60 | $D_f$=2.60 | $D_f$=2.80 |
| 2.71 | actual value | 1.14 | 1.14 | 1.14 | 1.16 | 1.14 |
| | measured value | 1.40 | 1.53 | 1.56 | 1.63 | 1.63 |
| | relative error | 22.71% | 34.03% | 36.92% | 40.60% | 42.84% |
| 2.37 | actual value | 1.06 | 1.07 | 1.08 | 1.23 | 1.23 |
| | measured value | 1.13 | 1.36 | 1.43 | 1.60 | 1.60 |
| | relative error | 6.84% | 27.29% | 32.26% | 30.35% | 29.54% |
| 2.15 | actual value | 1.02 | 1.01 | 1.02 | 1.27 | 1.28 |
| | measured value | 0.91 | 1.18 | 1.28 | 1.47 | 1.47 |
| | relative error | -10.55% | 16.85% | 25.30% | 16.05% | 14.80% |