# Peer review of "Numerical investigation on retrieval errors of mixing states of fractal black carbon aerosols using single-particle soot photometer based on Mie scattering and the effects on radiative forcing estimation"

_Atmospheric Measurement Techniques, 2023_

## Referee Comment (RC3)

Authors have brought to the attention of the scientific community the potential "flaw" in the current retrieval process for the Single Soot Photometers (SP2). They compared the current method, which is based on the spherical Mie core-shell assumption, with the closed-cell and coated aggregate models. It has been nicely illustrated by the authors what the differences would be in the $D_p/D_c$, $C_{sca}$, MAC, MSC, and RF if a closed-cell and coated aggregate model is used. In light of the increasing use of SP2 in atmospheric sciences, this paper is important, well-written and relevant for publication in this journal. It is a well-constructed study, the methodology is clearly explained, and reasonable conclusions are drawn. After addressing some comments mainly regarding the results and discussion, my recommendation is for publication.

1. The authors mostly provide the message that: "The measurement errors of mixing state have larger effect on the estimation accuracy of radiative forcing for heavily coated BC particles than that for thinly coated BC particles at both 1064 and 532 nm wavelengths."

   Generally, the authors report that discrepancies are primarily concentrated in heavily coated particles (for all parameters $D_p/D_c$, $C_{sca}$, MAC, MSC, and RF ). In my opinion, it is relevant to both thinly coated particles and heavily coated particles. When a fractal morphology is used to model BC in thinly coated particles (uniform coating around a BC fractal - closed cell model), it underestimates $D_p/D_c$ and $C_{sca}$ when compared to current Mie theory-based retrievals. In contrast, for heavily coated particles (where the BC fractal is completely enclosed in a spherical coating), the $D_p/D_c$ and $C_{sca}$ are overestimated compared to the current Mie theory-based estimates. In other words, it may be necessary to change the retrieval process depending on the type of mixture or age of the aerosols the SP2 measures. A major message from this research is that the community should adapt the current Mie core-shell retrieval according to the ageing stage of BC being measured.

2. What I found lacking was a discussion of comparisons between experimental and field applications. In light of the findings of this study, the next step would be to validate the $D_p/D_c$ derived from the actual SP2 measurements using the spherical Mie core-shell assumption and the closed-cell and coated aggregate models. There should also be a comparison between this measurement and maybe other chemical measurements that report the BC mass/volume fraction, such as filter measurements. In order for the findings of this study to be fully validated, the findings of a field or laboratory study are necessary (outlook).

3. As a follow-up to comment 2, the authors should mention previous experimental studies that have demonstrated how important it is to consider BC as a fractal aggregate. Until now, no one has presented any information regarding SP2.

   He, C., Liou, K.-N., Takano, Y., Zhang, R., Levy Zamora, M., Yang, P., Li, Q., and Leung, L. R.: Variation of the radiative properties during black carbon aging: theoretical and experimental intercomparison, Atmos. Chem. Phys., 15, 11967–11980, https://doi.org/10.5194/acp-15-11967-2015, 2015.

Romshoo, B., Pöhlker, M., Wiedensohler, A., Pfeifer, S., Saturno, J., Nowak, A., Ciupek, K., Quincey, P., Vasilatou, K., Ess, M. N., Gini, M., Eleftheriadis, K., Robins, C., Gaie-Levrel, F., and Müller, T.: Importance of size representation and morphology in modelling optical properties of black carbon: comparison between laboratory measurements and model simulations, Atmos. Meas. Tech., 15, 6965–6989, https://doi.org/10.5194/amt-15-6965-2022, 2022.

Forestieri, S. D., Helgestad, T. M., Lambe, A. T., Renbaum-Wolff, L., Lack, D. A., Massoli, P., Cross, E. S., Dubey, M. K., Mazzoleni, C., Olfert, J. S., Sedlacek III, A. J., Freedman, A., Davidovits, P., Onasch, T. B., and Cappa, C. D.: Measurement and modeling of the multiwavelength optical properties of uncoated flame-generated soot, Atmos. Chem. Phys., 18, 12141–12159, https://doi.org/10.5194/acp-18-12141-2018, 2018.

4. Figures 3, 4, show the value of Dp,v/Dc,v, which is the standard constant in each subfigure. It would be helpful to include the value of volume fraction in brackets in order to make it more relevant to those who do not use SP2, as well.

5. In Figure 5, it would be helpful to be able to see the slope in each sub-plot. By doing so, one would be able to determine how high or low the models are in relation to one another in terms of numbers.

6. There is a higher error in the calculation of the radiative forcing for thickly coated particles when Dp,v/Dc,v is higher. It would be interesting to see how the forcing values change for cases with Dp,v/Dc,v close to 1, i.e., thinly coated particles (as I discussed in the first comment).

7. Last but not least, a "beta" version of the retrieval code based on the closed-shell or coated aggregate model would be helpful to the community. It is possible for the authors to share the codes that they used in this study. However, this will only be a test version that would be developed in the future.

---

## Author Comment (AC1)

**RESPONSE LETTER (amt-2023-53)**

Title: Numerical investigation on measurement errors of mixing states of fractal black carbon aerosols using single-particle soot photometer and the effects on radiative forcing estimation

Dear Joshua Schwarz:

We have revised our manuscript based on your comments. The corrections and modifications have been included in the revised manuscript and the details are listed as follows. The responses are highlighted in blue font. The changes made in the revised manuscript are marked in red font.

I was happy to see this submission out focusing on improving interpretation of data types such as obtained with a single particle soot photometer (SP2). As a specialist with this instrument, I can clearly understand (and appreciate) the value of this to the community of SP2 users. The SP2 measures a few quantities on a per-particle basis relevant to determining mixing state. First, it provides the refractory black carbon (rBC) mass content of a particle (within some range of mass). This is based on an optical measurement of thermal emission, and is quite robust. Secondly, with appropriate analysis and setup, the instrument *measures* the total particle optical size – it detects a scattered light signal, and quantifies it. This is also valid only over some range of particle optical size, and is specific to the geometry of detection of the SP2. Third, some groups use the optical size of *only* the rBC portion of the particle (which can often be measured after the detection technique evaporates non-rBC material). Finally, inspection of the evolution of scattered light as a particle interacts with the SP2 laser provides another indication of internal mixing of materials with rBC. These measurements have been dealt with at length in the literature. After a measurement one can say: this particle had XX femtograms of rBC content, scattered as much light as a YY nm-diameter polystyrene latex sphere (PSL) into the SP2 scattering detector, and showed (or did not) evidence of shrinking during heating. These measurements have statistical and systematic errors associated with them, but are independent of Mie or any other theory of light scattering from particles. Now, the point at which this submitted manuscript becomes relevant is in the interpretation of those measured

quantities. With knowledge about the amount of rBC mass and the total particle scattering signal, how can we interpret these quantities to infer conclusions about the amount of non-rBC material and its impacts on light absorption?

Presently, the paper is presented as though dealing with "SP2 measurement errors". This is not the case. Rather, it deals with assessing Mie-theory inadequacies for complex aerosols (which are highly relevant to SP2 analyses). This is a more general topic of interest to a wider slice of the community than SP2 users/interpreters, but has been addressed often in the literature in the broad sense. Hence, I recommend maintaining the focus on the SP2 community.

Response:

Thanks a lot for reviewing our manuscript and for your appreciation! The highly condensed introduction of SP2 you gave is very meaningful for us to deepen our understanding of how SP2 works and improve our research. We have responded to all these constructive comments point by point and modified related descriptions in the revised manuscript.

Broad comments:

The focus on "measurement errors" should be adjusted to more correctly address "interpretation effects". This is important because there is nothing wrong with the measurements that are published, and which remain valid independent of the interpretation method. Note that this manuscript does not actually determine "errors" (which would require measurements for comparison) - rather it determines differences between different optical calculations applied to interpretations. I think this was reasonably summed up in our 2015 paper on measurements and interpretation with a humidified SP2, which I quote here in part to suggest some additional references that should be added to the paper and considered. The final sentence of this quoted section speaks directly to the value of the manuscript under consideration:

"The SP2 user community often relies on Mie theory to interpret SP2-measured particle scattering cross-section and rBC mass content for the amount of non-rBC material internally

mixed with the particles [Schwarz et al., 2008]. Although there is mild experimental support for the SP2 determination of coating thickness via Mie-theory assuming a shell-and-core morphology [Laborde et al, 2012], recent results hint at the uncertainties associated with this approach. Scarnato et al. [2013] used discrete dipole approximations as well as Mie theory to explore morphology effects on scattering and absorption of bare and internally mixed BC. They show (their Figure 3) that there can be considerable differences (~2X) between the exact numerical methods and the Mie-theory approximation for light-scattering at 1000 nm wavelength (near the SP2 wavelength of 1064 nm). Moteki et al. [2014] included comparison of SP2 light-scattering observations from near core-shell morphologies of BC coated with oleic acid via vapor deposition. They observed a bias up to 40% compared to Mie shell-and-core theory estimates constrained by total particle mass, rBC mass, and the (known) index of refraction of the oleic acid. Exploration of the validity of Mie theory approximations is beyond the scope of this manuscript, but is clearly relevant."

Response:

Thank you very much for the valuable comments and suggestions! We fully agree that there is nothing wrong in the original data measured by SP2, the errors concerned and investigated in our study occur when the optical equivalent diameter ($D_p$) of coated black carbon is retrieved based on original data using different interpretation methods and models, that is the core-shell model and the fractal model.

For the sake of accuracy, all the "measurement error" have been modified to "retrieval error", the title of the manuscript also has been modified, and the descriptions of recommended references have been added to the introduction in the revised manuscript:

"Numerical investigation on retrieval errors of mixing states of fractal black carbon aerosols using single-particle soot photometer based on Mie scattering and the effects on radiative forcing estimation"

"Mie scattering theory, which assumes that coated BC particle has a concentric core-shell structure consisting of coating sphere and BC sphere, is usually employed to retrieve the optical equivalent diameter of the coated BC based on differential scattering cross-section measured by SP2 (Schwarz et al., 2008; Kompalli et al., 2021). Finally, the particle size ratio of the whole particle to the BC core ($D_p/D_c$) can be obtained. Experimental results obtained by Laborde et

al. (2012) showed that Mie scattering theory can be employed to retrieve the coating thickness of aged BC particles based on SP2 measurements. However, comparisons conducted by Scarnato et al. (2013) revealed that the scattering and absorption of the internally mixed BC calculated by the discrete dipole approximation (DDA) and the Mie scattering theory may be considerably different at 1000 nm (close to the 1064 nm used by SP2). Moteki et al. (2014) emphasized that the optical properties simulated by Mie theory deviate from the SP2 observations as much as 40% affected by the total particle mass, the rBC mass, and the refractive index of oleic acid coating. Schwarz et al. (2015) also proposed that when the SP2 was used to quantify the water-uptake of BC particles coated by ammonium sulfate, the uncertainty of SP2 measured results was mainly caused by the significant deviations in predicting SP2 scattering properties of BC particles using Mie scattering. In summary, it can be deduced that there are unavoidable retrieval errors in $D_p/D_c$ because the core-shell model used in the retrieval of optical equivalent particle size $D_p$ does not conform to the non-spherical complex morphology of the coated BC particles."

Response:

   Thanks for this constructive comment! We re-conducted the retrieval of mixing states $D_p/D_c$ of coated soot aerosols based on partial scattering cross-section corresponding to the specific detection geometry of SP2 rather than total scattering cross-section during our revision, which is more in line with the measurement principle of SP2. The Figures and Tables in the manuscript vary more or less, and the discoveries and conclusions are re-drawn. All these modifications are included in the revised manuscript.

A lot of value would be added to the paper for the SP2 community if, in addition to addressing this error, it was made easier for SP2 users to use the results of the numerical simulations. I'm suggesting that the authors consider including lookup tables that could be used by SP2 users (rather than the mie-theory look ups that are currently more commonly used). The format of these tables would be up to the authors, but I'd suggest something similar to what we use: a dimension for the rBC mass content (or volume-equivalent diameter for an assumed density) and a dimension for the amount of internally mixed material (a mass or volume ratio, again

with an assumed density for the internally mixed material). In our lookups we also vary the real index of refraction of the internally mixed material as a third dimension, but this would likely be overkill here. Each entry of the table would then provide the partial scattering cross-section, as would be measured with the the LEO approach with the SP2. Different tables for the different fractal dimensions of the rBC could be used, or that could be added as an additional dimension of the table. Additional tables with absorption information would then complete the set that would be commonly used by the community. I don't think this is necessary for publication, but would represent a great contribution and example for how future numerical studies could be more impactful, if the authors are willing to publish it. Note, too, that this would strengthen the relevance of the paper for AMT.

Response:

Thanks for this constructive suggestion! We cannot agree more a lookup table or database as you mentioned is very necessary and meaningful for the SP2 users and the community. In our opinion, such a lookup table must include a large amount of calculated differential scattering cross-section corresponding to SP2 measurements for coated soot particles, morphological models for coated soot particles at different aging stage such as thinly coated model, partially coated model, and thickly coated models should be considered, different micro-physical parameters with wide value range and small step size such as fractal dimension, monomer size, monomer number, soot volume fraction, complex refractive index also should be taken in to consideration. In this manuscript, we only tentatively explore the errors in retrieved $D_p/D_c$ of coated soot aerosols caused by core-shell morphological assumption using closed-cell model and coated aggregate model with several micro-physical parameters. Therefore, a public lookup table based on this manuscript will not be helpful to SP2 users and even can be misleading to some degree. After sufficient exploration as mentioned above, then a comprehensive lookup table will be more meaningful for the community.

Specific comments:

The authors make the point that aging leads to more compact particles. I think it would be good to also cite China et al, "Morphology and Mixing State of Aged Soot Particles at a Remote Marine Free-tropospheric Site: Implications to Optical Properties", 2015 for context here (with their conclusion that Mie theory is within 12% of DDA for the older rBC-containing aerosol).

Response:

Thank you very much for the suggestion! The descriptions of recommended reference have been added to the introduction in the revised manuscript:

"During the aging process in the atmospheric environment, BC will be coated by other species, and their aggregate morphology tends to be more compact (China et al., 2013). Combined observation and simulation carried out by China et al. (2015) showed that Mie calculations provide reasonable approximations for compact soot above remote marine clouds, and the distinction of radiative forcing estimated using Mie theory and using discrete dipole approximation is within 12% for a high surface albedo."

Line 44 – there has also been a fair number of publications using the SP2 fraction of rBC-containing particles that show evidence of being internally mixed (often referred to as "thinly vs thickly coated rBC".

Response:

Thanks a lot for this comment! For the sake of accuracy, the related description have been modified in the revised manuscript as follows:

"Currently, the mixing states of rBC-containing particles are mainly characterized using the following methods: the particle diameter ratio of the whole particles to the BC core ($D_p/D_c$), the coating thickness (CT), the SP2 measured numerical fractions of thinly and thickly coated rBC, and the mass ratio of the coating material to the BC core ($M_R$)."

Line 67: this connects to my first broad comment. SP2 does not measure $D_p/D_c$, and does not have unavoidable errors in the LEO scattering measurement. The Mie theory interpretations do not destroy the information in the quantities measured by the SP2, they only transform them into different spaces (coating thickness or $D_p/D_c$), which can still be used to infer the original

observed quantities, and allow reinterpretation with another optical model. Similarly, the table headings titled "SP2 retrieved core-shell models" – these are Mie-theory core-shell models. (Note that we have also used RDG to interpret SP2 data… Mie theory is not tied to the SP2 or vice versa.)

Response:

Thanks a lot for the constructive comments! As in the response to the first broad comment, we have modified the description "measurement error" to "retrieval error" in the revised manuscript. In addition, Line 67 and the headings of Tables 3 and 4 in the original manuscript have been modified as follows:

"In summary, it can be deduced that there are unavoidable errors in the retrieved $D_p/D_c$ based on Mie theory because the core-shell model used in the retrieval of optical equivalent particle size $D_p$ does not conform to the nonspherical complex morphology of the coated BC particles. At present, the retrieval error in $D_p/D_c$ of coated BC based on SP2 measurement results is difficult to be quantified directly through experimental investigations. Nevertheless, the rapid developments of both morphology modeling and optical simulation of coated BC particles provide an investigative strategy for evaluating the retrieval accuracy of $D_p/D_c$."

"**Table 3.** The SFE values of both the core-shell models used to interpret the SP2 measurements and the fractal soot models at 1064 nm."

"**Table 4.** The SFE values of both the core-shell models used to interpret the SP2 measurements and the fractal soot models at 532 nm."

To summarize – this is a very promising entry that could provide a lot of value to the SP2 community. Making sure that the calculations are as relevant as possible to the geometry of the SP2 is one requirement. Another is correcting the association of interpretation differences to instrumental error. The authors also have the opportunity to provide a data set that I suspect would be broadly used in SP2-science.

Response:

Thank you for your valuable suggestions. We have responded to the comments point by point and revised the manuscript. We sincerely invite you to review our manuscript again.

**References**

China, S., Mazzoleni, C., Gorkowski, K., Aiken, A. C., and Dubey, M. K.: Morphology and mixing state of individual freshly emitted wildfire carbonaceous particles, Nat Commun, 4, 2122, 10.1038/ncomms3122, 2013.

Kompalli, S. K., Babu, S. N. S., Moorthy, K. K., Satheesh, S. K., Gogoi, M. M., Nair, V. S., Jayachandran, V. N., Liu, D. T., Flynn, M. J., and Coe, H.: Mixing state of refractory black carbon aerosol in the South Asian outflow over the northern Indian Ocean during winter, Atmospheric Chemistry and Physics, 21, 9173-9199, 10.5194/acp-21-9173-2021, 2021.

Laborde, M., Schnaiter, M., Linke, C., Saathoff, H., Naumann, K. H., Mohler, O., Berlenz, S., Wagner, U., Taylor, J. W., Liu, D., Flynn, M., Allan, J. D., Coe, H., Heimerl, K., Dahlkotter, F., Weinzierl, B., Wollny, A. G., Zanatta, M., Cozic, J., Laj, P., Hitzenberger, R., Schwarz, J. P., and Gysel, M.: Single Particle Soot Photometer intercomparison at the AIDA chamber, Atmospheric Measurement Techniques, 5, 3077-3097, 10.5194/amt-5-3077-2012, 2012.

Moteki, N., Kondo, Y., and Adachi, K.: Identification by single-particle soot photometer of black carbon particles attached to other particles: Laboratory experiments and ground observations in Tokyo, Journal of Geophysical Research-Atmospheres, 119, 1031-1043, 10.1002/2013jd020655, 2014.

Scarnato, B. V., Vahidinia, S., Richard, D. T., and Kirchstetter, T. W.: Effects of internal mixing and aggregate morphology on optical properties of black carbon using a discrete dipole approximation model, Atmospheric Chemistry and Physics, 13, 5089-5101, 10.5194/acp-13-5089-2013, 2013.

Schwarz, J. P., Perring, A. E., Markovic, M. Z., Gao, R. S., Ohata, S., Langridge, J., Law, D., McLaughlin, R., and Fahey, D. W.: Technique and theoretical approach for quantifying the hygroscopicity of black-carbon-containing aerosol using a single particle soot photometer, J. Aerosol. Sci., 81, 110-126, 10.1016/j.jaerosci.2014.11.009, 2015.

Schwarz, J. P., Spackman, J. R., Fahey, D. W., Gao, R. S., Lohmann, U., Stier, P., Watts, L. A., Thomson, D. S., Lack, D. A., Pfister, L., Mahoney, M. J., Baumgardner, D., Wilson, J. C., and Reeves, J. M.: Coatings and their enhancement of black carbon light absorption in the tropical atmosphere, Journal of Geophysical Research-Atmospheres, 113, 10, 10.1029/2007jd009042, 2008.

Furthermore, other detailed revisions are listed below.

| LOCATION | REVISED MANUSCRIPT | ORIGINAL MANUSCRIPT |
|---|---|---|
| Abstract, paragraph 1 | deviated from the real morphology | deviated the real morphology |
| | references | reference |
| | the diameter of BC core ($D_c$) is | the diameter of BC core ($D_c$) are |
| | the mixing state ($D_p/D_c$) | mixing state ($D_p/D_c$) |
| | aspects | aspect |
| | at most particle sizes | for most particle sizes |
| Introduction, paragraph 1 | acts | act |
| Introduction, paragraph 3 | mixing state of each single BC particle | mixing state of a single BC particle |
| | at first | first |
| Introduction, paragraph 5 | observation | observed |
| | provide insight into the possible errors | provide insight of the possible errors |
| Section 2.1, paragraph 2 | organic | organics |
| Section 2.1, paragraph 3 | ranges | range |
| | relationships | relationship |
| Section 2.3, paragraph 1 | the scattering signal of each coated BC particle | the scattering signal of coated BC particles |
| | the coated BC | coated BC |
| Section 2.3, paragraph 2 | with the value of $D_c$ | and the value of $D_c$ |

| | | |
|---|---|---|
| Section 3.3, paragraph 3 | effects | effect |
| | of the coated-aggregate model | on the coated-aggregate model |
| Section 3.4, paragraph 1 | have significant impacts | have a significant impact |
| | The SP2 retrieves | SP2 measurement |
| Section 3.4, paragraph 2 | effects | effect |

---

## Author Comment (AC3)

**RESPONSE LETTER (amt-2023-53)**

Title: Numerical investigation on measurement errors of mixing states of fractal black carbon aerosols using single-particle soot photometer and the effects on radiative forcing estimation

Dear reviewer:

We have revised our manuscript based on your comments. The corrections and modifications have been included in the revised manuscript and the details are listed as follows. The responses are highlighted in blue font. The changes made in the revised manuscript are marked in red font.

General comments:

This study aims to evaluate the uncertainty of the amount of coating material derived from the measured scattering-cross section by SP2 depending on the particle shape assumption. In particular, the authors focused on evaluations of the systematic error due to the assumption of the shell-core particle shape model, which has been widely used but unrealistic for real-world BC-containing particles. The research focus is of significance to every SP2 user. However, I found some issues which must be addressed (corrected) before considering publication.

Response:

Thanks a lot for reviewing our manuscript and all these constructive comments. We have responded to the comments point by point and modified related descriptions in the revised manuscript.

1.The "closed-cell model" adopted for thinly-coated BC looks very different from the real-world BC-containing particles. I can't agree with using such a fictitious model as a "reference" for quantifying the error of the conventional shell-core model. In my opinion, the authors should use another more realistic shape model (e.g., according to TEM images) for thinly-coated BC. The distance between neighboring BC monomers, which artificially depends on the BC volume fraction in the closed-cell model, can be a critical factor in determining the optical properties of BC aggregates because the multipole interaction between monomers strongly depends on the distance, as implied by, for example, Mackowski 1995

https://doi.org/10.1364/AO.34.003535. If the authors are using "the closed-cell model", the authors should show evidence of its accuracy.

Response:

Thanks a lot for your valuable comments!

We employed the closed-cell model to simulate the thinly coated BC particle at the early stage of atmospheric aging, because a number of previous studies have chosen this model to represent coated BC particles, and simulation results of optical properties were in good agreement with the corresponding results measured from laboratory experiments and field observations. He et al. (2016) used closed-cell model to represent black carbon coated by sulfuric acid, results showed that optical properties such as scattering cross-section, absorption cross-section, and asymmetry factor of closed-cell models can closely match the laboratory measurements. Romshoo et al. (2022) conducted comparisons of optical properties between laboratory measurements and model simulations of closed-cell models, they also found the modeled light absorption coefficient, the single-scattering albedo, and the mass absorption cross-section are in good consistencies with the measurements.

Furthermore, the closed-cell model consists of numerous core-shell structures of soot and coating, the coating structure is fixed, and the effects of fractal parameters of coated particles will be more evident. However, the calculated optical properties will be affected by the distance between neighboring BC cores caused by the change in the volume fraction of BC, as the reviewer pointed out. We agree that realistic shape models based on TEM images, like models developed by Luo et al. (2023), will be more meaningful, and this is our further direction for effort. With the assistance of such models, the effects of the complex coating structure of soot on the measurements of SP2 are expected to be revealed.

2. The authors seem to (implicitly) assume the scattering cross-section measured by the SP2 as if it is the total scattering cross-section. In fact, the scattering cross-section measured by the SP2 is only a small fraction of the "4pi str integral" of the differential scattering cross-section. The authors should explain this fact and evaluate the maximum error caused by the assumption.

Response:

Thank you very much for pointing this out! We re-conducted the retrieval of mixing states of coated soot aerosols based on differential scattering cross-section corresponding to the specific detection geometry of SP2 rather than total scattering cross-section during our revision, which is more in line with the measurement principle of SP2. The Figures and Tables in the manuscript vary more or less, and the discoveries and conclusions are re-drawn. All these modifications are included in the revised manuscript.

Specific comments:

L11: "measured mixing state"

The meaning of the "mixing state" could depend on the context and it is not obvious if it is a well-defined physical quantity (such as "volume"). I recommend using "volume of coating" or "volume fraction of coating" for clarity throughout the manuscript.

Response:

Thanks a lot for this valuable suggestion! We have modified the related descriptions to "diameter ratio of coated particle to BC core" or directly "$D_p/D_c$" in the revised manuscript. The L11 in the original manuscript has been modified as follows:

"The mixing state of black carbon (BC) aerosols, that is the diameter ratio of coated particle to BC core ($D_p/D_c$), can be retrieved by the single-particle soot photometer (SP2). However, the retrieved $D_p/D_c$ contains errors, because the core-shell model and Mie scattering calculation are normally employed in the retrieval principle of SP2 and the spherical core-shell structure seriously deviated from the real morphology of coated BC."

L13: "thinly and heavily"

"thinly and thickly" or "lightly and heavily" sounds more accurate.

Response:

Thank you very much for this suggestion! We have modified "heavily" to "thickly" in the revised manuscript. The L13 in the original manuscript has been modified as follows:

"In this study, fractal models are constructed to represent thinly and thickly coated BC particles for optical simulations, …"

L15: "the diameter of BC core ($D_c$)"

Is it volume-equivalent diameter? Please clarify.

Response:

Thanks for pointing this out! The diameter of the BC core ($D_c$) is the volume equivalent diameter, which can be derived from the mass equivalent diameter measured by SP2 with the assistance of soot density $1.8 g/cm^3$. The L15 in the original manuscript has been modified as follows for clarity:

"…, and the volume equivalent diameter of BC core ($D_c$) is the same for fractal and spherical models."

L25: "is considered to be the second most important factor affecting global warming after carbon dioxide (Zhang et al., 2021)"

I'm not sure how this statement is still supported by recent climate research. In the IPCC AR6 report, methane was considered to have a larger positive effective radiative forcing than BC.

Response:

Thank you for the valuable comment! We have carefully read the IPCC AR6 report, for the sake of rigor, the L25 in the original manuscript has been modified as follows:

"Black carbon (BC) produced from the incomplete combustion of biomass and fossil fuels is considered to be an important contributor to global warming (Zhang et al., 2021)."

L34: "affects the vertical diffusion" suppress the vertical diffusion?

Response:

Thanks a lot for your comments! Black carbon aerosols can significantly weaken the diffusion and dilution of pollutants by heating the atmosphere, which in turn worsens air quality. In order to be more precise, the "affects" have been modified by "suppresses".

"…, which further suppresses the vertical diffusion of air pollutants, enhances haze events, harms human health, and reduces atmospheric visibility (Huang et al., 2018)."

L57: "single-particle soot photometer" You should use "SP2" which has already been defined.

Response:

Thanks for this suggestion! The L57 in the original manuscript has been modified as follows:

"The SP2 measures the mass and differential scattering cross-section of each single BC particle based on the combination of laser-induced incandescent light technology and light scattering measurement technology."

L59: "The scattering cross-section of the BC particle can be rapidly retrieved based on the measurement results of the scattering signal detectors (Schwarz et al., 2006)." The methods for retrieval of the scattering cross-section (integrated over the solid angle of light collection) using SP2 were introduced by Gao et al. 2007 AST (with position-sensitive detector) and Moteki and Kondo 2008 JAS (without position-sensitive detector). Please refer to at least one of these papers here.

Response:

Thank you for the constructive suggestion. We have added related descriptions and the above two references in the revised manuscript:

"The SP2 measures the mass and differential scattering cross-section of each single BC particle based on the combination of laser-induced incandescent light technology and light scattering measurement technology. In the optical cavity, when a coated BC particle vertically passes through the high-energy laser beam at a wavelength of 1064 nm, there are two scattering signal detectors collect the scattering signal over certain solid angles at forward and backward directions, respectively (Schwarz et al., 2006). Gao et al. (2007) developed the leading-edgeonly (LEO) fit method to deal with the collected scattering signal for SP2 with position-sensitive detector, the undisturbed leading edge of the scattering signal was employed to construct a Gaussian scattering function, then the Gaussian function can be used to determine the differential scattering cross-section of the BC-containing particle. On the other hand, for SP2 without position-sensitive detector, Moteki and Kondo (2008) proposed to measure the time-dependent differential scattering cross-section ($\Delta C_{sca}(t)$) as each particle flowing across the Gaussian laser beam, and the differential scattering cross-section for coated BC particles can be further obtained. In general, the coated particles rapidly absorb the laser energy, the coating is heated to vaporization at first, and then the refractory BC (rBC) is heated and emits incandescent light (Zhao et al., 2021)."

L62: "The intensity of the incandescent light signal is proportional to the mass of rBC" This statement sounds like oversimplifying the truth. The linear proportionality between the LII signal and BC mass is only valid under a limited condition of BC size and LII detection wavelengths.

Pease see Moteki and Kondo 2010 AS&T https://doi.org/10.1080/02786826.2010.484450.

Response:

Thank you very much for this valuable comments! In order to be more precise, we have modified the description in the revised manuscript as follows:

"At 1064 nm wavelength, when the rBC with particle masses range between the lower and upper detection limits of SP2, the intensity of the incandescent light signal is proportional to the mass of rBC, and the volume equivalent particle diameter of rBC can be obtained based on the preset density (1.80g/cm$^3$) (Moteki and Kondo, 2010)."

L217: "The distribution of retrieved results of mixing states for single-particle with different fractal dimensions over the entire particle size range is shown in Figure 5, and the filling width represents the probability distribution of retrieved $D_p/D_c$." I'm not sure if this can be regarded as a "probability" distribution. I guess from the context that each violin plot in Figure 5 shows a histogram of the retrieved $D_p/D_c$ values for uniformly-sampled $D_{c,v}$ ordinate. Is my guess correct? If so, Figure 5 is just a different plot of the data shown in Figures 3 and 4?

Response:

Thank you for the meaningful comment!

Indeed, Figures 3, 4, and 5 are drawn based on the same database of calculated optical properties of closed-cell model, coated aggregate model, and core-shell model. However, these figures are drawn for the analyses of retrieved $D_p/D_c$ from different aspects. Figure 3 and 4 are plotted to reveal the variation of retrieved $D_p/D_c$ and retrieval error with volume equivalent diameter of soot core, while Figure 5 shows the variation of retrieved $D_p/D_c$ with the preset $D_{p,v}/D_{c,v}$, and all the coated BC particles with different core diameter are considered at each $D_{p,v}/D_{c,v}$. Essentially, Figure 5 is a diagram of probability distribution or frequency distribution, but it should be noted that the value of $D_{p,v}/D_{c,v}$ is not sampled uniformly, because the volume fraction of soot core was preset and $D_{p,v}/D_{c,v}$ is derived correspondingly. The violin plot is statistically significant, and the filling width represents the frequency of retrieved $D_p/D_c$. A wider violin plot indicates that the corresponding retrieved result of $D_p/D_c$ accounts for a larger proportion of all the retrieved results.

**References**

Gao, R. S., Schwarz, J. P., Kelly, K. K., Fahey, D. W., Watts, L. A., Thompson, T. L., Spackman, J. R., Slowik, J. G., Cross, E. S., Han, J. H., Davidovits, P., Onasch, T. B., and Worsnop, D. R.: A novel method for estimating light-scattering properties of soot aerosols using a modified single-particle soot photometer, Aerosol Science and Technology, 41, 125-135, 10.1080/02786820601118398, 2007.

He, C. L., Takano, Y., Liou, K. N., Yang, P., Li, Q. B., and Mackowski, D. W.: Intercomparison of the GOS approach, superposition T-matrix method, and laboratory measurements for black carbon optical properties during aging, Journal of Quantitative Spectroscopy & Radiative Transfer, 184, 287-296, 10.1016/j.jqsrt.2016.08.004, 2016.

Huang, X., Wang, Z., and Ding, A.: Impact of Aerosol-PBL Interaction on Haze Pollution: Multiyear Observational Evidences in North China, Geophysical Research Letters, 45, 8596-8603, 10.1029/2018gl079239, 2018.

Luo, J., Li, Z. Q., Qiu, J. B., Zhang, Y., Fan, C., Li, L., Wu, H. L., Zhou, P., Li, K. T., and Zhang, Q. X.: The Simulated Source Apportionment of Light Absorbing Aerosols: Effects of Microphysical Properties of Partially-Coated Black Carbon, Journal of Geophysical Research-Atmospheres, 128, 20, 10.1029/2022jd037291, 2023.

Moteki, N. and Kondo, Y.: Method to measure time-dependent scattering cross sections of particles evaporating in a laser beam, J. Aerosol. Sci., 39, 348-364, 10.1016/j.jaerosci.2007.12.002, 2008.

Moteki, N. and Kondo, Y.: Dependence of Laser-Induced Incandescence on Physical Properties of Black Carbon Aerosols: Measurements and Theoretical Interpretation, Aerosol Science and Technology, 44, 663-675, 10.1080/02786826.2010.484450, 2010.

Romshoo, B., Pohlker, M., Wiedensohler, A., Pfeifer, S., Saturno, J., Nowak, A., Ciupek, K., Quincey, P., Vasilatou, K., Ess, M., Gini, M., Eleftheriadis, K., Robins, C., Gaie-Levrel, F., and Muller, T.: Importance of size representation and morphology in modelling opticalproperties of black carbon: comparison between laboratory measurements andmodel simulations, Atmospheric Measurement Techniques, 15, 6965-6989, 10.5194/amt-15-6965-2022, 2022.

Schwarz, J. P., Gao, R. S., Fahey, D. W., Thomson, D. S., Watts, L. A., Wilson, J. C., Reeves, J. M., Darbeheshti, M., Baumgardner, D. G., Kok, G. L., Chung, S. H., Schulz, M., Hendricks, J., Lauer, A., Karcher, B., Slowik, J. G., Rosenlof, K. H., Thompson, T. L., Langford, A. O., Loewenstein, M., and Aikin, K. C.: Single-particle measurements of midlatitude black carbon and light-scattering aerosols from the boundary layer to the lower stratosphere, Journal of Geophysical Research-Atmospheres, 111, 15, 10.1029/2006jd007076, 2006.

Zhang, X., Mao, M., Chen, H., and Tang, S.: The single scattering albedo Angstrom exponent of black carbon with brown coatings, Journal of Quantitative Spectroscopy and Radiative Transfer, 259, 10.1016/j.jqsrt.2020.107429, 2021.

Zhao, G., Tan, T. Y., Zhu, Y. S., Hu, M., and Zhao, C. S.: Method to quantify black carbon aerosol light absorption enhancement with a mixing state index, Atmospheric Chemistry and Physics, 21, 18055-18063, 10.5194/acp-21-18055-2021, 2021.

Furthermore, other detailed revisions are listed below.

| LOCATION | REVISED MANUSCRIPT | ORIGINAL MANUSCRIPT |
|---|---|---|
| Abstract, paragraph 1 | deviated from the real morphology | deviated the real morphology |
| | references | reference |
| | the diameter of BC core ($D_c$) is | the diameter of BC core ($D_c$) are |
| | the mixing state ($D_p/D_c$) | mixing state ($D_p/D_c$) |
| | aspects | aspect |
| | at most particle sizes | for most particle sizes |
| Introduction, paragraph 1 | acts | act |
| Introduction, paragraph 3 | mixing state of each single BC particle | mixing state of a single BC particle |
| | at first | first |
| Introduction, paragraph 5 | observation | observed |
| | provide insight into the possible errors | provide insight of the possible errors |
| Section 2.1, paragraph 2 | organic | organics |
| Section 2.1, paragraph 3 | ranges | range |
| | relationships | relationship |
| Section 2.3, paragraph 1 | the scattering signal of each coated BC particle | the scattering signal of coated BC particles |
| | the coated BC | coated BC |
| Section 2.3, paragraph 2 | with the value of $D_c$ | and the value of $D_c$ |

| Section 3.3, paragraph 3 | effects | effect |
|---|---|---|
| | of the coated-aggregate model | on the coated-aggregate model |
| Section 3.4, paragraph 1 | have significant impacts | have a significant impact |
| | The SP2 retrieves | SP2 measurement |
| Section 3.4, paragraph 2 | effects | effect |

---

## Author Comment (AC4)

**RESPONSE LETTER (amt-2023-53)**

Title: Numerical investigation on measurement errors of mixing states of fractal black carbon aerosols using single-particle soot photometer and the effects on radiative forcing estimation

Dear reviewer:

  We have revised our manuscript based on your comments. The corrections and modifications have been included in the revised manuscript and the details are listed as follows. The responses are highlighted in blue font. The changes made in the revised manuscript are marked in red font.

  Authors have brought to the attention of the scientific community the potential "flaw" in the current retrieval process for the Single Soot Photometers (SP2). They compared the current method, which is based on the spherical Mie core-shell assumption, with the closed-cell and coated aggregate models. It has been nicely illustrated by the authors what the differences would be in the $D_p/D_c$, $C_{sca}$, MAC, MSC, and RF if a closed-cell and coated aggregate model is used. In light of the increasing use of SP2 in atmospheric sciences, this paper is important, well-written and relevant for publication in this journal. It is a well-constructed study, the methodology is clearly explained, and reasonable conclusions are drawn. After addressing some comments mainly regarding the results and discussion, my recommendation is for publication.

Response:

  Thanks a lot for reviewing our manuscript and all these constructive comments. We have responded to the comments point by point and modified related descriptions in the revised manuscript.

1. The authors mostly provide the message that: "The measurement errors of mixing state have larger effect on the estimation accuracy of radiative forcing for heavily coated BC particles than that for thinly coated BC particles at both 1064 and 532 nm wavelengths."

Generally, the authors report that discrepancies are primarily concentrated in heavily coated particles (for all parameters $D_p/D_c$, $C_{sca}$, MAC, MSC, and RF). In my opinion, it is relevant to both thinly coated particles and heavily coated particles. When a fractal morphology is used to model BC in thinly coated particles (uniform coating around a BC fractal - closed cell model), it underestimates $D_p/D_c$ and Csca when compared to current Mie theory-based retrievals. In contrast, for heavily coated particles (where the BC fractal is completely enclosed in a spherical coating), the $D_p/D_c$ and Csca are overestimated compared to the current Mie theory-based estimates. In other words, it may be necessary to change the retrieval process depending on the type of mixture or age of the aerosols the SP2 measures. A major message from this research is that the community should adapt the current Mie core-shell retrieval according to the ageing stage of BC being measured.

Response:

Thanks a lot for your valuable comments!

We re-conducted the retrieval of mixing states of coated soot aerosols based on differential scattering cross-section corresponding to the detectors of SP2 rather than total scattering cross-section during our revision, which is more in line with the measurement principle of SP2. Results showed that the measurement errors of the mixing state have larger effects on the estimation accuracy of radiative forcing for thinly coated BC particles than that for heavily coated BC particles at both 1064 and 532 nm wavelengths. Since the retrieved values of $D_p/D_c$ generally decrease with the preset volume equivalent diameter of the soot core, the measured mixing states based on Mie theory can be underestimated or overestimated for coated particles with different morphologies, as shown in Figures 3 and 4 in the revised manuscript.

We agree with you that the retrieval progress for thinly and heavily coated soot particles are necessary to be improved respectively. Furthermore, it should be noted that the micro parameters and coating structures of soot aerosols are various at different regions and times because of photochemical aging and hygroscopic growth, these aging processes should be systematically considered from numerical aspects and the corresponding measurement errors of BC mixing states should also be noticed by the community. We have modified related descriptions in the revised manuscript as follows:

"The retrieval errors of mixing state have larger effects on the estimation accuracy of radiative forcing for thinly coated BC particles than that for thickly coated BC particles at both 1064 and 532 nm wavelengths."

2. What I found lacking was a discussion of comparisons between experimental and field applications. In light of the findings of this study, the next step would be to validate the $D_p/D_c$ derived from the actual SP2 measurements using the spherical Mie core-shell assumption and the closed-cell and coated aggregate models. There should also be a comparison between this measurement and maybe other chemical measurements that report the BC mass/volume fraction, such as filter measurements. In order for the findings of this study to be fully validated, the findings of a field or laboratory study are necessary (outlook).

Response:

Thanks for this constructive suggestion!

We agree that experimental validation is meaningful for both making our work more complete and for researchers using SP2. However, a comprehensive field or laboratory study includes aerosol particle sampling, electron microscope observation, chemical analysis, particle diameter characterization, SP2 measurement, and optical modeling, and it is difficult for us to organize such observations at the present stage. Furthermore, there are still several technological challenges unresolved, for example, the conversion between the measured electrical mobility diameter or aerodynamic diameter and the volume equivalent diameter for coated BC particles. Our previous study (Liu et al., 2020) found that the relationship between electrical mobility diameter and volume equivalent diameter is very complicated for BC, the unsuitable conversion parameters will also bring huge uncertainties to the measurements. On the other hand, even though these two kinds of fractal models we employed are typical to some degree, they cannot be representative of all types of atmospheric BC particles, and more morphological models should be considered in our research framework. Therefore, we focus on the retrieval errors from numerical aspects only in this study. And we have added the necessary descriptions as future works in the revised manuscript as follows:

"This study is a pilot work for the characterization of possible retrieval errors in mixing states of coated soot aerosols using SP2 and Mie theory. For future work, more morphological models that are suitable for modeling the microstructure of coated soot aerosols should be considered, and unified parameterization schemes of retrieval errors are much needed. Furthermore, comprehensive field or laboratory studies for the validation of the possible errors in mixing states are also future directions worth the effort."

3. As a follow-up to comment 2, the authors should mention previous experimental studies that have demonstrated how important it is to consider BC as a fractal aggregate. Until now, no one has presented any information regarding SP2.

He, C., Liou, K.-N., Takano, Y., Zhang, R., Levy Zamora, M., Yang, P., Li, Q., and Leung, L. R.: Variation of the radiative properties during black carbon aging: theoretical and experimental intercomparison, Atmos. Chem. Phys., 15, 11967–11980, h.ps://doi.org/10.5194/acp-15-11967-2015, 2015.

Romshoo, B., Pöhlker, M., Wiedensohler, A., Pfeifer, S., Saturno, J., Nowak, A., Ciupek, K., Quincey, P., Vasilatou, K., Ess, M. N., Gini, M., ElePheriadis, K., Robins, C., GaieLevrel, F., and Müller, T.: Importance of size representation and morphology in modelling optical properties of black carbon: comparison between laboratory measurements and model simulations, Atmos. Meas. Tech., 15, 6965–6989, h.ps://doi.org/10.5194/amt-15-6965-2022, 2022.

Forestieri, S. D., Helgestad, T. M., Lambe, A. T., Renbaum-Wolff, L., Lack, D. A., Massoli, P., Cross, E. S., Dubey, M. K., Mazzoleni, C., Olfert, J. S., Sedlacek III, A. J., Freedman, A., Davidovits, P., Onasch, T. B., and Cappa, C. D.: Measurement and modeling of the multiwavelength optical properties of uncoated flame-generated soot, Atmos. Chem. Phys., 18, 12141–12159, h.ps://doi.org/10.5194/acp-18-12141-2018, 2018.

Response:

Thank you for the valuable comments and recommendations!

We have added related descriptions in the introduction, and the above three references have been cited in the revised manuscript.

"In short, abundant models and numerical simulation algorithms of BC provide convenience for accurate calculation of BC optical properties and also create an effective way

to quantify the possible errors of the mixing states $D_p/D_c$ of BC retrieved by SP2. In addition, there have been a number of experimental studies demonstrating the importance of considering BC as fractional aggregates. He et al. (2015) developed three different fractal aggregate models to simulate three typical evolution stages of BC particles and demonstrated that the dynamic aging process and the fractal shape should be considered for accurate estimations of the radiation effects. Romshoo et al. (2022) calculated the optical properties of three fractal particle models, results showed that the aggregate representation performs well in modeling the light absorption coefficient, the single-scattering albedo, and the mass absorption cross-section for laboratory-generated BC particles with mobility diameters larger than 100 nm. The fractal aggregate model was selected by Forestieri et al. (2018) to model uncoated soot particles, optical calculation results were used to compare with experimentally measured optical properties at multiwavelength, they emphasized that the sphere model and Mie theory widely used in climate models may lead to obvious underprediction in absorption of BC. All these studies have demonstrated the excellent performances of fractal aggregates in the optical modeling of black carbon. However, the fractal particle models have not been employed in the SP2 retrieval research, and the retrieval errors of the soot mixing state caused by morphological model selection also have not been evaluated."

4. Figures 3, 4, show the value of $D_{p,v}/D_{c,v}$, which is the standard constant in each subfigure. It would be helpful to include the value of volume fraction in brackets in order to make it more relevant to those who do not use SP2, as well.

Response:

Thanks for this valuable suggestion! The corresponding values of volume fractions have been added in parentheses after the values of $D_{p,v}/D_{c,v}$ in Figures 3 and 4 in the revised manuscript. In addition, Figures 3 and 4 have been redrawn due to the modification of the retrieval process. We have modified the figures in the revised manuscript as follows:

[Figure]

**Figure 3**. Retrieved mixing state ($D_p/D_c$) and relative error ($RE$) as functions of volume equivalent diameter ($D_{c,v}$) for thinly coated BC particles with different $D_f$ and $D_{p,v}/D_{c,v}$. The colored lines stand for retrieved $D_p/D_c$ and the colored bars stand for $RE$.

[Figure]

**Figure 4**. Retrieved mixing state ($D_p/D_c$) and relative error ($RE$) as functions of volume equivalent diameter ($D_{c,v}$) for heavily coated BC particles with different $D_f$ and $D_{p,v}/D_{c,v}$. (a) Coated-aggregates with $D_f = 2.60$; (b) Coated-aggregates with $D_f = 2.80$. The colored lines stand for retrieved $D_p/D_c$ and the colored bars stand for $RE$.

5. In Figure 5, it would be helpful to be able to see the slope in each sub-plot. By doing so, one would be able to determine how high or low the models are in relation to one another in terms of numbers.

Response:

Thanks for this suggestion! In the course of our research, the volume fractions were preset, and the values of $D_{p,v}/D_{c,v}$ were derived subsequently. Therefore, the horizontal axis $D_{p,v}/D_{c,v}$ in Figure 5 is not equally spaced and we draw this figure using independent scatters, then the slope in each sub-plot may make little sense. However, it can still be clearly observed that the averaged $D_p/D_c$ slightly decreases with the decrease of $D_{p,v}/D_{c,v}$, except for thinly coated soot particles with $D_{p,v}/D_{c,v} = 1.36, 1.13$, because of the loss of some retrieved values of $D_p/D_c$.

6. There is a higher error in the calculation of the radiative forcing for thickly coated particles when $D_{p,v}/D_{c,v}$ is higher. It would be interesting to see how the forcing values change for cases with $D_{p,v}/D_{c,v}$ close to 1, i.e., thinly coated particles (as I discussed in the first comment).

Response:

Thank you for the valuable comment! We set the values of volume fractions of soot core in coated particles in this study, and the volume fractions can be related to $D_{p,v}/D_{c,v}$ one by one. The $D_{p,v}/D_{c,v}$ roughly covers the retrieved results $D_p/D_c$ using SP2 in previous literature, and the $D_{p,v}/D_{c,v}$ is close to 1(1.13) when the volume fraction is 0.70. However, the inherent differences between the optical properties of fractal models calculated using MSTM and optical properties of fractal models calculated using Mie theory result in the loss of retrieved values of mixing state, especially for particles with $D_{p,v}/D_{c,v}$ close to 1. Therefore, these cases were not considered for the evaluation of radiative forcing. We are developing complex morphological models more realistic than closed-cell models for thinly coated particles, and we expect to be able to investigate the radiative forcing of thinly coated particles in detail based on the new morphological model.

7. Last but not least, a "beta" version of the retrieval code based on the closed-shell or coated aggregate model would be helpful to the community. It is possible for the authors to share the codes that they used in this study. However, this will only be a test version that would be developed in the future.

Response:

Thanks for this constructive suggestion! A complete and user-friendly code package of the whole calculation and retrieval progress is meaningful for the researchers of SP2 observation and optical simulation. However, our current study conducted optical calculation and retrieval based on several independent scripts written based on Matlab, and such a code package has not been developed yet. On the other hand, as further research of this manuscript, we are investigating the effects of hygroscopic growth and photochemical aging on the SP2 measurements on the basis of these scripts,  so we would to publish our scripts and package when we finish the ongoing study in the future.

**References**

Forestieri, S. D., Helgestad, T. M., Lambe, A. T., Renbaum-Wolff, L., Lack, D. A., Massoli, P., Cross, E. S., Dubey, M. K., Mazzoleni, C., Olfert, J. S., Sedlacek, A. J., Freedman, A., Davidovits, P., Onasch, T. B., and Cappa, C. D.: Measurement and modeling of the multiwavelength optical properties of uncoated flame-generated soot, Atmospheric Chemistry and Physics, 18, 12141-12159, 10.5194/acp-18-12141-2018, 2018.

He, C., Liou, K. N., Takano, Y., Zhang, R., Zamora, M. L., Yang, P., Li, Q., and Leung, L. R.: Variation of the radiative properties during black carbon aging: theoretical and experimental intercomparison, Atmospheric Chemistry and Physics, 15, 11967-11980, 10.5194/acp-15-11967-2015, 2015.

Liu, J., Zhang, Q. X., Wang, J. J., and Zhang, Y. M.: Light scattering matrix for soot aerosol: Comparisons between experimental measurements and numerical simulations, Journal of Quantitative Spectroscopy & Radiative Transfer, 246, 14, 10.1016/j.jqsrt.2020.106946, 2020.

Romshoo, B., Pohlker, M., Wiedensohler, A., Pfeifer, S., Saturno, J., Nowak, A., Ciupek, K., Quincey, P., Vasilatou, K., Ess, M., Gini, M., Eleftheriadis, K., Robins, C., Gaie-Levrel, F.,

and Muller, T.: Importance of size representation and morphology in modeling opticalproperties of black carbon: comparison between laboratory measurements and model simulations, Atmospheric Measurement Techniques, 15, 6965-6989, 10.5194/amt-15-6965-2022, 2022.

Furthermore, other detailed revisions are listed below.

| LOCATION | REVISED MANUSCRIPT | ORIGINAL MANUSCRIPT |
|---|---|---|
| Abstract, paragraph 1 | deviated from the real morphology | deviated the real morphology |
| | references | reference |
| | the diameter of BC core ($D_c$) is | the diameter of BC core ($D_c$) are |
| | the mixing state ($D_p/D_c$) | mixing state ($D_p/D_c$) |
| | aspects | aspect |
| | at most particle sizes | for most particle sizes |
| Introduction, paragraph 1 | acts | act |
| Introduction, paragraph 3 | mixing state of each single BC particle | mixing state of a single BC particle |
| | at first | first |
| Introduction, paragraph 5 | observation | observed |
| | provide insight into the possible errors | provide insight of the possible errors |
| Section 2.1, paragraph 2 | organic | organics |
| Section 2.1, paragraph 3 | ranges | range |
| | relationships | relationship |

| | | |
|---|---|---|
| Section 2.3, paragraph 1 | the scattering signal of each coated BC particle | the scattering signal of coated BC particles |
| | the coated BC | coated BC |
| Section 2.3, paragraph 2 | with the value of $D_c$ | and the value of $D_c$ |
| Section 3.3, paragraph 3 | effects | effect |
| | of the coated-aggregate model | on the coated-aggregate model |
| Section 3.4, paragraph 1 | have significant impacts | have a significant impact |
| | The SP2 retrieves | SP2 measurement |
| Section 3.4, paragraph 2 | effects | effect |

---

## Author Response (AR2)

**RESPONSE LETTER (amt-2023-53)**

Title: Numerical investigation on measurement errors of mixing states of fractal black carbon aerosols using single-particle soot photometer and the effects on radiative forcing estimation

Dear Joshua Schwarz:

We have revised our manuscript based on your comments. The corrections and modifications have been included in the revised manuscript and the details are listed as follows. The responses are highlighted in blue font. The changes made in the revised manuscript are marked in red font.

Comments:

Unfortunately, I think another revision is necessary, as it appears that your new calculations don't combine forward and back-scattering. The SP2 uses a single intra-cavity laser - meaning that a single scattering detector measures the combined forward- and back-scattering from each particle. If you were already combining them for your main figures, this can merely be corrected/clarified in the text. In the Supplemental figure 1, however, it doesn't make sense to separate forward and back scattering, as the SP2 is not capable of doing this. Here better to just show combine forward/back into a single scalar measure of scattering for the comparison.

Response:

Thanks a lot for reviewing our revised manuscript again and for the valuable constructive comments. We also noticed the point you mentioned above during our retrieval study.

As shown in the last column of Figure 1, the dashed lines represent the sum of forward and backward differential scattering cross-sections for different fractal models (closed-cell model and coated-aggregate model) with all the preset morphology parameters at fixed BC core diameters ($D_c$). The curves indicate the variation of the sum of the forward and backward differential scattering cross sections with coated particle diameter $D_p$ for the spherical core-shell model with fixed $D_c$. The horizontal coordinate corresponding to the point where the dashed line and the curve intersect is the retrieved result of the optical equivalent $D_p$. It can be seen from Figure 1 that the curves corresponding to the sum of forward and backward

differential scattering cross-sections become more and more flat with the increase of $D_c$, which means that less optical equivalent $D_p$ can be retrieved based on Mie theory. Therefore, we infer that the mixing state of some coated soot particles may be ignored by SP2 in the laboratory and field observations.

Therefore, we further explore the performance of the BC mixing state retrieve based on forward and backward scattering separately. Through the comprehensive analyses of all the simulated optical properties, we found that compared to retrieval using backward scattering and a combination of both forward and backward scattering, the retrieval using forward scattering can avoid the coated BC particles being missed by SP2 to the greatest extent. For example, for BC particles with $D_c$=371.3 nm, the mixing states of soot particles missed by using forward scattering is half of that missed by using a combination of both forward and backward scattering. Therefore, we think retrieving the mixing state using forward scattering only may be the most efficient for coated BC particles which can be roughly represented by the closed-cell model and the coated aggregate model.

Our results are meaningful in improving the optical cavity structure of the SP2 instrument. However, it should be noted that we only focused on the closed-cell model and coated-aggregate model, mixing state retrieval performance comparisons of retrieval using forward scattering, backward scattering, and a combination of both forward and backward scattering should be further conducted in future based on more kinds of complex morphological models of coated soot particles.

[Figure]

**Figure 1**. The variations of differential scattering cross-sections with coated particle diameter $D_p$ of core-shell models for soot core diameter $D_c$=185.7 nm, 267.8 nm, 371.3 nm, 467.8 nm. The dashed lines indicate the differential scattering cross-sections of the fractal model with different morphology parameters. The first column represents the forward differential scattering cross-sections, the second column represents the backward differential scattering cross-sections, and the third column represents the sum of the differential scattering cross-sections integrated in both directions.

For the sake of rigor, related descriptions in the L169, L205, and L375 in the firstly revised manuscript have been modified as follows in the second version of the revised manuscript:

"The two scattering signal detectors of the SP2 are distributed at forward scattering and backward scattering directions to simultaneously measure the forward and backward scattering of each particle, which have view angles ($\theta$) range in 13-77° and 103-167°, respectively (Wu et al., 2023)."

"When the combination of forward and backward differential scattering cross-sections is employed to retrieve BC mixing state, $D_p/D_c$ for a noteworthy amount of coated soot particles cannot be retrieved based on Mie scattering theory due to the different optical properties between spherical and fractal models, which means that the mixing states of some coated soot particles may be ignored. Therefore, the retrieval performances of the BC mixing state based on forward and backward scattering are further explored, separately. Comprehensive comparisons showed that the retrieval based on forward scattering can avoid the coated BC particles being missed by SP2 to the greatest extent. Therefore, the retrieved results of mixing states based on forward scattering are selected for further discussion in the following sections. **Figure 3** shows the retrieval results and relative errors (*RE*) of mixing states of thinly coated soot under different fractal dimensions ($D_f$) and volume equivalent particle diameter ratio shell/core ($D_{p,v}/D_{c,v}$), the colored lines and bars stand for retrieved $D_p/D_c$ and *RE*, respectively."

"For future work, more morphological models that are suitable for modeling the microstructure of coated soot aerosols should be considered, the retrieval performance of mixing state based on forward scattering, backward scattering, and their combination should be explored, and unified parameterization schemes of retrieval errors are much needed. Furthermore, comprehensive field or laboratory studies for the validation of the possible errors in mixing states are also future directions worth the effort. "